# Perfect fluids

Jan de Boer[1], Jelle Hartong[1], Niels A. Obers[2], Watse Sybesma[3] and Stefan Vandoren[3]

**1** Institute for Theoretical Physics and Delta Institute for Theoretical Physics,
University of Amsterdam, Science Park 904, 1098 XH Amsterdam, The Netherlands
**2** The Niels Bohr Institute, Copenhagen University,
Blegdamsvej 17, DK-2100 Copenhagen Ø, Denmark
**3** Institute for Theoretical Physics and Center for Extreme Matter and Emergent Phenomena,
Utrecht University, 3508 TD Utrecht, The Netherlands

## Abstract

We present a systematic treatment of perfect fluids with translation and rotation symmetry, which is also applicable in the absence of any type of boost symmetry. It involves introducing a fluid variable, the *kinetic mass density*, which is needed to define the most general energy-momentum tensor for perfect fluids. Our analysis leads to corrections to the Euler equations for perfect fluids that might be observable in hydrodynamic fluid experiments. We also derive new expressions for the speed of sound in perfect fluids that reduce to the known perfect fluid models when boost symmetry is present. Our framework can also be adapted to (non-relativistic) scale invariant fluids with critical exponent $z$. We show that perfect fluids cannot have Schrödinger symmetry unless $z = 2$. For generic values of $z$ there can be fluids with Lifshitz symmetry, and as a concrete example, we work out in detail the thermodynamics and fluid description of an ideal gas of Lifshitz particles and compute the speed of sound for the classical and quantum Lifshitz gases.


doi:10.21468/SciPostPhys.5.1.003

# 1  Introduction

Perfect fluids are fluids that at rest are completely described in terms of their energy density $\mathscr{E}$ and pressure $P$. The energy-momentum tensor in the rest frame of a fluid element then takes a particularly simple form with the energy and pressure on the diagonal elements (assuming rotational symmetry). One can then describe the fluid in a moving frame by introducing Lorentz or Galilei boost velocities $v^i$, and impose the conservation laws of the energy-momentum tensor due to translational symmetry in space and time. Similarly there may be additional conserved currents, expressing the conservation of particle number, electric charge, baryon number etc. Supplemented with an equation of state, one derives a consistent set of equations that determines the system. The perfect fluid description has found many applications in physical systems, both in the relativistic and non-relativistic cases. For a general treatment, see e.g. the classic textbook by Landau and Lifshitz, volume 6 [1].

Many systems, however, do not have boost symmetry. In particular the boost symmetry can be broken as soon as there is a preferred reference frame, a medium with respect to which the fluid moves. Let us recall that boost symmetry allows us to relate all inertial frames, i.e. observers moving with respect to each other with constant velocity. In the Galilean (non-relativistic) and Lorentzian (relativistic) cases, such boosts look like

$$\text{Galilei boost}: \qquad \vec{x}^{\,'} = \vec{x} - \vec{v}\,t\,, \qquad t' = t\,, \tag{1}$$

$$\text{Lorentz boost}: \qquad \vec{x}^{\,'} = \gamma(\vec{x} - \vec{v}\,t)\,, \qquad t' = \gamma\left(t - \frac{\vec{v}\cdot\vec{x}}{c^2}\right)\,, \tag{2}$$

where $\vec{v}$ is the boost velocity between the different inertial frames, and the Lorentz factor is given by $\gamma = (1 - v^2/c^2)^{-1/2}$. At low velocities $v^2 \ll c^2$, the Lorentz boost reduces to the Galilei boost, and this is the non-relativistic limit. In the absence of boost symmetry, one has to study the fluid in each inertial frame, characterized by $\vec{v}$. Hence we will give a fluid description where the velocity $\vec{v}$ is a parameter that cannot be set to zero by boost symmetry. The case $\vec{v} = 0$ is of course the rest frame.

There are many systems in which boost symmetry is broken. A well-known example from biology is the fluid description of bird flocks moving through the air (the medium). Another example is an electron gas moving in a lattice of atoms, where the electron-phonon interactions break boost invariance. At small length scales, the medium itself can break translation and rotation symmetry, and as a consequence also boost symmetry. At larger distances, e.g. at

distances larger than the lattice spacing, translation symmetry can effectively be restored, but boost symmetry may remain absent. Throughout this article, this will be our working assumption. In other words, we will assume that observers at rest see an isotropic and homogeneous fluid, but without boost symmetry. Another class of examples that we will discuss in detail are the Lifshitz fluids. On top of translation and rotations, they obey scale symmetry characterized by a dynamical exponent, but they do not have any boost symmetry.

The main aim of our work is to give a systematic analysis of fluids in which there is translational and rotational symmetry, but no boost symmetry. Certain aspects of hydrodynamics without boost symmetry have, often in particular examples, been studied before, such as e.g. [2] and references therein, but most often this is done in the rest frame, without keeping track of all the velocity (frame)-dependence. Here we aim for a unified framework and treatment that applies to both systems with and without boost, and that holds in any frame. In this paper, we focus on perfect fluids. A more complete treatment of general fluids and hydrodynamics without boost symmetry will be given in accompanying work [3,4]. A general consequence of the absence of boost symmetry is that different inertial frames are no longer related by boost transformations, and our fluid description therefore has to include the boost velocity to keep track of the different inertial frames. As we will show, this can easily be done by treating the boost velocity as a chemical potential. A further consequence, in contradistinction with the relativistic case, is that the energy-momentum tensor is no longer symmetric, i.e. $T^{0i} \neq T^{i0}$. This will lead to a new fluid variable that we call the *kinetic mass density*, $\rho$, which has the dimension of a mass-density[1]. For the standard Galilean invariant fluids, this kinetic mass density is proportional (up to factors of $c$) to the particle density $n$ times mass of the fluid particle $m$, while for Lorentz-invariant fluids it is proportional to the enthalpy density $\mathscr{E} + P$. In the absence of boosts, this relation will be broken, and one treats $\rho$ as an independent fluid variable. As a consequence, the equation of state involves one more parameter to completely specify the system, so that for example the energy density $\mathscr{E}(s, n, \rho)$ not only depends on entropy density $s$ and particle density $n$, but also on the kinetic mass density $\rho$. Alternatively, this means that the pressure $P(T, \mu, v^2)$ has an extra non-trivial dependence on the fluid velocity.

The first main result of this paper is to show how the kinetic mass density enters the formulation of perfect fluids, and to find an explicit form of the energy-momentum tensor in the absence of boost symmetry. As a consistency check, imposing boost symmetry on our new energy-momentum tensor, we can reproduce the known relativistic and Galilean cases, since these symmetries dictate a relation between $\rho$ and different variables. Moreover, we also find the form of a perfect Carrollian, or ultra-relativistic, fluid, which can be obtained by taking $c \to 0$ in the Lorentzian case. Therefore, our analysis provides a unified framework that can be applied to all these cases separately. In general, as we will show, the kinetic mass density can be computed from the pressure of the fluid in a boosted frame.

An important characteristic quantity that can be computed for a perfect fluid is the speed of sound $v_s$. It can be obtained from the fluctuation analysis of the energy-momentum tensor. Since our energy-momentum tensor is modified due to broken boost invariance, the analysis of the sound modes needs to be reinvestigated. Furthermore, the sound speed needs to be computed for arbitrary background fluid velocities, since one cannot rely on the Doppler effect that relates the sound speeds in different frames. This analysis is performed in this paper and is the second of our main results. A third important general result is a no-go theorem for perfect Schrödinger fluids and furthermore an extension to the case of fluids with hyperscaling violation and the presence of a charge anomalous dimension.

---

[1]It is interesting to note that the quantity $\rho$ was previously seen in the context of Lifshitz holography, namely it appeared in the holographic perfect fluid description of a new class of moving Lifshitz black branes [5] in the Einstein-Proca-dilaton model. This is yet another example of the power of holography to predict new aspects of hydrodynamics.

We illustrate our general description of perfect fluids by studying an ideal gas of Lifshitz particles. In general, critical systems with Lifshitz symmetry have scale invariance of the form

$$t \to \Lambda^z t , \quad \vec{x} \to \Lambda \vec{x} . \tag{3}$$

Here $z$ is the so-called dynamical critical exponent. One typically has $z \geq 1$ and not necessarily any type of boost symmetry[2]. Using our general framework, we derive the speed of sound for an ideal gas of Lifshitz particles. We furthermore review several thermodynamical quantities such as heat capacities, equations of state, and for the quantum Lifshitz gas, the conditions for Bose-condensation to take place. In the rest frame, some of the results can be found at various places in the literature, but we also extend it to other frames. The results of this paper are thus expected to be of relevance to systems with critical scaling behavior set by a dynamical exponent $z$, including strongly coupled quantum critical systems.

The organization of this paper is as follows. In Section 2, we present our general description of perfect fluids starting from a thermodynamical partition function that includes the fluid velocity as a chemical potential. As a consistency check, we rederive the known cases by imposing boost symmetry, and we discuss models with Lifshitz and Carroll symmetries. In Section 3, we derive expressions for the speed of sound, in the rest frame, and present the case for general background fluid velocity $\vec{v}_0$ in Appendix A. In Section 4, we study an ideal gas model of Lifshitz particles, with classical, bosonic, and fermionic statistics. As a proof of principle we compute the speed of sound for those models. We end with an outlook in Section 5.

## 2 Perfect fluids

We will give a universal definition of a perfect fluid (on flat space-time) assuming only time and space translational symmetries as well as rotational symmetries. In particular we will not assume that there are boost symmetries. These may or may not be broken. We will discuss how this universal definition incorporates all known cases of relativistic and non-relativistic (Galilean and Carrollian) perfect fluids as well as more novel fluids that have no boost symmetries such as Lifshitz perfect fluids. The discussion in this paper holds in arbitrary number of spatial dimensions $d$, and we label them by $i = 1, ..., d$.

### 2.1 Thermodynamics and kinetic mass density

Let us start with a thermodynamic system described by a partition function $\mathscr{Z}$ in a grand canonical ensemble that depends on temperature $T$, volume $V$, a chemical potential for variable particle number $\mu$ and (importantly) a velocity vector $v^i$. We assume thermal equilibrium and treat the velocity as a chemical potential constant in spacetime, to describe the fluids in different inertial frames.

In quantum mechanical language, the grand canonical partition function and thermal density matrix can be written as ($\beta = (k_B T)^{-1}$),

$$\mathscr{Z} = \text{Tr}\left( e^{-\beta(\hat{H} - \mu\hat{N} - \vec{v}\cdot\hat{\vec{P}})} \right) , \qquad \rho_\beta \equiv \frac{e^{-\beta(\hat{H} - \mu\hat{N} - \vec{v}\cdot\hat{\vec{P}})}}{\mathscr{Z}} , \tag{4}$$

---

[2]Boosts are possible if one also adds a $U(1)$ current with the scaling weight $z - 2$. Such systems are then said to have Schrödinger symmetry. They can be studied using the traditional fluid dynamics, and so we will not pay any particular attention to them. Moreover, we will show in later sections that no Schrödinger perfect fluids exist with $z \neq 2$.

with mutually commuting operators, the Hamiltonian $\hat{H}$, particle number $\hat{N}$ and (total) momentum $\hat{\vec{P}}$ with conjugate position operator $\hat{\vec{Q}}$. The velocity $\vec{v}$ is also the average total velocity and we have the relations

$$\vec{v} = \langle \dot{\hat{\vec{Q}}} \rangle = \frac{i}{\hbar} \langle [\hat{H}, \hat{\vec{Q}}] \rangle = \left\langle \frac{\partial \hat{H}}{\partial \hat{\vec{P}}} \right\rangle . \tag{5}$$

The last equal sign can also directly be shown using a complete set of momentum eigenstates and the defining relation for the expectation value of an operator in a thermal state, $\langle \hat{O} \rangle = \mathrm{Tr}(\rho_\beta \hat{O})$.

From the grand potential $\Omega$, defined as

$$\Omega = -k_B T \log \mathscr{Z}(T, V, \mu, v^i) , \tag{6}$$

we can compute entropy $S$, pressure $P$, and the expectation values of particle number $N$ and momentum $P_i$ via

$$d\Omega = -S dT - P dV - P_i dv^i - N d\mu . \tag{7}$$

We assume that the grand potential is an extensive quantity, from which it follows that $\Omega = -PV$. Furthermore, one can derive the expression $\Omega = E - TS - v^i P_i - \mu N$, where $E$ is the energy, $E = \langle \hat{H} \rangle = \mathrm{Tr}(\rho_\beta \hat{H})$. It then follows that

$$E = TS - PV + v^i P_i + \mu N , \qquad dE = T dS - P dV + v^i dP_i + \mu dN . \tag{8}$$

In terms of the densities

$$\mathscr{E} = \frac{E}{V} , \qquad \mathscr{P}_i = \frac{P_i}{V} , \qquad n = \frac{N}{V} , \qquad s = \frac{S}{V} , \tag{9}$$

we easily derive

$$\mathscr{E} = Ts - P + v^i \mathscr{P}_i + \mu n , \qquad d\mathscr{E} = T ds + v^i d\mathscr{P}_i + \mu dn . \tag{10}$$

We have introduced a chemical potential $v^i$ with the dimensions of velocity whose conjugate variable is the total momentum density of the system $\mathscr{P}_i$. Since there is only one vector and we assume rotational symmetry, we can say without loss of generality that

$$\mathscr{P}_i = \rho v^i , \tag{11}$$

for some function $\rho$ that has the dimensions of a mass density. Since this function plays an important role in this paper, we will introduce a name for it, and call it the *kinetic mass density*.

The pressure $P$ can be seen as function of the temperature, chemical potential and velocity, $P(T, \mu, v^2)$, as follows from

$$dP = s dT + n d\mu + \frac{1}{2} \rho dv^2 . \tag{12}$$

So another way to compute the kinetic mass density is by

$$\rho(T, \mu, v^2) = 2 \left( \frac{\partial P}{\partial v^2} \right)_{T,\mu} . \tag{13}$$

The kinetic mass density is a new thermodynamic quantity[3]. For boost invariant cases, it will reduce to other known quantities, such as the usual mass density $\rho = mn$ in the non-relativistic Galilean case, or it will relate to energy and pressure in the relativistic case, as

---

[3]It has appeared in the literature before, such as [2] (see Eqn.(459) in Section 5.4.5 in this reference, where is was called "equilibrium susceptibility"), but only its $v$-independent part. In [2], some aspects of non-relativistic hydrodynamics are also studied, but no systematic development of the velocity dependence was given. In particular, the relation between the kinetic mass density and the pressure given in (13) has not appeared before in the literature.

we will see later. In general, in the absence of boost symmetry, a new equation of state for $\rho$ needs to be provided to determine the system, or, as a different way to state the same fact, one can compute the velocity dependent terms in the pressure function and take the derivative as in (13). The kinetic mass density expresses the relation between momentum and velocity. In Galilean single particle dynamics, the relation between momentum and velocity is of course a simple mass factor, and in the Lorentzian case, it is the mass times the Lorentz factor. In general, this relation can be more complicated and defines the kinetic mass. An explicit example will be given in Section 4, when we discuss the Lifshitz ideal gas.

We have here written the pressure $P$ and the kinetic mass density $\rho$ as functions of $(T, \mu, v^2)$. The entropy and particle number density $s$ and $n$ are then derived quantities and also depend on $(T, \mu, v^2)$, via e.g. $s = (\partial P / \partial T)_{\mu, v^2}$. Similarly for the energy density $\mathscr{E}$. We can use these relations to switch variables and write the pressure as a function of different variables. Later, in Section 3 for instance, we will write the pressure as a function of $P(\mathscr{E}, n, v^2)$ or we will choose the energy density as a function $\mathscr{E}(s, n, v^2)$. The thermodynamic relations allow us to do so at our convenience. But as usual, one must specify carefully which quantities are kept constant upon partial differentiation. What is important is that the equilibrium system with rotation symmetry can be described in terms of three functions. For the perfect fluid, these can be chosen to be energy, pressure, and kinetic mass density.

This completes the thermodynamical analysis of systems without boost invariance. The discussion is generally applicable to any system in thermal equilibrium. We now turn to the fluid description, in which we eventually perturb away from global equilibrium by varying all quantities in space and time. We do this for perfect fluids, i.e. maintaining local equilibrium, whose energy-momentum tensor we define in the rest or laboratory frame. In [3, 4] we consider deviations from local equilibrium due to transport effects such as viscosity and conductivity.

## 2.2 A new energy-momentum tensor

As mentioned in the introduction, we assume that the fluid enjoys time $H$ and space $P_i$ translation as well as rotational $J_{ij}$ invariance. Furthermore we will assume that there is a global $U(1)$ symmetry generated by a charge $Q$. The corresponding conserved quantity can be electric charge or particle number like baryon number. We don't need to specify this in subsequent analysis. The conserved currents that generate these symmetries are $T^\mu{}_0$ for $H$, $T^\mu{}_i$ for $P_i$, $x^i T^\mu{}_j - x^j T^\mu{}_i$ for $J_{ij}$ and $J^\mu$ for $Q$ and the conservation equations read $\partial_\mu T^\mu{}_\nu = 0$, $T^i{}_j = T^j{}_i$ and $\partial_\mu J^\mu = 0$. For example, if the fluid admits a Lagrangian description we assume that there exist improvements of the Noether currents such that the above statements are true. At this stage we will not assume the presence of scale symmetries. This will be done later.

It should be clear that, even though we use $(\mu, \nu)$ indices, there is no Lorentz symmetry. Similarly, we cannot raise or lower indices with some spacetime metric, so the natural position of indices on the energy-momentum tensor is one upper and one lower, denoted as $T^\mu{}_\nu$. As usual, this matrix will have one negative eigenvalue, namely minus the energy.

We will use a static coordinate system $(x^0 = t, x^i)$, the laboratory or LAB frame, in which an observer only moves in time along $\partial_0$. This means that the observer is at rest with respect to the effective medium in which the fluid flows. We assume that the space-time symmetries are generated by the Hermitian operators $\hat{H} = i\partial_0$, $\hat{P}_i = -i\partial_i$ and $\hat{J}_{ij} = -i(x^i\partial_j - x^j\partial_i)$. All commutators vanish except the ones between $\hat{P}_i$ and $\hat{J}_{ij}$,

$$[\hat{J}_{ij}, \hat{P}_k] = i\delta_{ik}\hat{P}_j - i\delta_{jk}\hat{P}_i \,. \tag{14}$$

Classically, the generators are realized by

$$H = -\int_V \mathrm{d}^d x \, T^0{}_0(x), \tag{15}$$

$$P_i = \int_V \mathrm{d}^d x \, T^0{}_i(x), \tag{16}$$

$$J_{ij} = \int_V \mathrm{d}^d x \left( x^i T^0{}_j(x) - x^j T^0{}_i \right), \tag{17}$$

$$Q = \int_V \mathrm{d}^d x \, J^0(x). \tag{18}$$

The currents $T^\mu{}_\nu$ and $J^\mu$ transform in representations of the group generated by $H$, $P_i$, $J_{ij}$ and $Q$.

   We will now define what we mean by a perfect fluid in the LAB frame. The time components of the energy $T^\mu{}_0$, momentum $T^\mu{}_i$ and charge $J^\mu$ currents are equal to minus the total energy density $\mathscr{E}$, the total momentum density $\mathscr{P}_i$ and the charge density $n$, respectively. The spatial components of these same currents are equal to the fluxes plus terms involving the pressure. In the LAB frame the charges $\mathscr{E}$, $\mathscr{P}_i$ and $n$ flow with velocity $v^i$. We thus define a perfect fluid to have the following energy-momentum tensor and $U(1)$ current

$$T^0{}_0 = -\mathscr{E}, \qquad T^0{}_j = \mathscr{P}_j, \qquad T^i{}_0 = -(\mathscr{E}+P)v^i, \qquad T^i{}_j = P\delta^i_j + v^i\mathscr{P}_j, \tag{19}$$
$$J^0 = n, \qquad J^i = nv^i. \tag{20}$$

We have added pressure in the usual way. The flow of energy is $(\mathscr{E}+P)v^i$ as a result of work done by the fluid. The stress is given by $P\delta^i_j + v^i\mathscr{P}_j$ due to pressure and momentum flow. If there is no $U(1)$ current, then the form of the energy-momentum tensor (19) still holds. Notice further that this is not the most general form for the energy-momentum tensor compatible with the symmetries. In general one can have five scalar quantities appearing in the energy-momentum tensor, and similarly we could have a different scalar in the $J^i$ component than in the $J^0$. However, in thermodynamic equilibrium all charges move with the same average velocity. This requirement reduces the number of free functions in the energy-momentum tensor from five down to three, and to one in the $U(1)$ current.

   What we mean by saying that the charges $\mathscr{E}$, $\mathscr{P}_i$ and $n$ flow with velocity $v^i$ is that for constant velocity $v^i$ (but with all other quantities $\mathscr{E}$, $\rho$ and $n$ functions of space and time) we can go to a moving coordinate system in which all the fluxes are zero and in which the charges are: $\mathscr{P}_i$ for momentum, $n$ for charge density and $\mathscr{E} - v^i\mathscr{P}_i$, which equals $Ts + \mu n - P$ and which we refer to as the internal energy, for the energy current. The coordinate transformation that sets the fluxes to zero and for which the time components of the energy $T^\mu{}_0$, momentum $T^\mu{}_i$ and charge $J^\mu$ currents are equal to minus the internal energy $\mathscr{E} - v^i\mathscr{P}_i$, total momentum $\mathscr{P}_i$ and charge $n$ density is simply given by the coordinate transformation

$$x^i = x'^i + v^i t', \qquad t = t', \tag{21}$$

and takes the form of a Galilean boost (note that this is in general not a symmetry of the system). In other words for fluids with a constant velocity $v^i$ in the primed frame ($x'^0 = t', x'^i$)

the energy-momentum tensor and $U(1)$ current take the form[4]:

$$T'^0{}_0 = -\left(\mathscr{E} - v^i \mathscr{P}_i\right), \qquad T'^0{}_j = \mathscr{P}_j, \qquad T'^i{}_0 = 0, \qquad T'^i{}_j = P\delta^i_j, \tag{22}$$

$$J'^0 = n, \qquad J'^i = 0. \tag{23}$$

Let us denote the internal energy $\tilde{\mathscr{E}} \equiv \mathscr{E} - v^i \mathscr{P}_i$. We can write the thermodynamic relations (10) as[5]

$$\tilde{\mathscr{E}} = Ts + \mu n - P, \qquad \mathrm{d}\tilde{\mathscr{E}} = T\mathrm{d}s + \mu\mathrm{d}n - \frac{1}{2}\rho\mathrm{d}v^2, \tag{24}$$

where we used the definition of $\rho$, i.e. $\mathscr{P}_i = \rho v^i$. From this one derives another relation for the kinetic mass density, namely

$$\rho(s, n, v^2) = -2\left(\frac{\partial\tilde{\mathscr{E}}}{\partial v^2}\right)_{s,n}. \tag{25}$$

We can also trade the entropy density for energy density and write the mass density as a function $\rho(\tilde{\mathscr{E}}, n, v^2)$. Similarly, we can write the pressure as a function $P(\tilde{\mathscr{E}}, n, v^2)$ which we will use in the next section.

One may wonder how it is possible that we see a momentum but no momentum flux in these coordinates. To see better what is happening let us consider the equation of momentum conservation $\partial_\mu T^\mu{}_j = 0$ which reads in LAB frame components

$$\left(\partial_0 + v^i\partial_i\right)\mathscr{P}_j + \partial_j P + \mathscr{P}_j\partial_i v^i = 0. \tag{26}$$

For constant velocity in the primed frame this becomes

$$\partial'_0\mathscr{P}_j + \partial_j P = 0, \tag{27}$$

where $\partial'_0 = \partial_{t'}$. Hence the momentum flux for constant $v^i$ can be transformed away and the only contribution to the force $\partial'_0\mathscr{P}_j$ is the negative gradient of the pressure. In other words we do not need momentum flux in the primed system because the time $t'$ rate of change of the momentum is fully determined by the pressure only.

## 2.3 Entropy current and Euler equation

The conservation equations $\partial_\mu T^\mu{}_\nu = 0$ are given by

$$0 = \left(\partial_0 + v^i\partial_i\right)\mathscr{E} + \left(\mathscr{E} + P\right)\partial_i v^i + v^i\partial_i P, \tag{28}$$

$$0 = \left(\partial_0 + v^i\partial_i\right)\mathscr{P}_j + \partial_j P + \mathscr{P}_j\partial_i v^i. \tag{29}$$

By contracting the second equation with $v^j$ and using this to eliminate $v^i\partial_i P$ from the first equation we obtain

$$\left(\partial_0 + v^i\partial_i\right)\mathscr{E} - v^j\left(\partial_0 + v^i\partial_i\right)\mathscr{P}_j + \left(\mathscr{E} + P - v^j\mathscr{P}_j\right)\partial_i v^i = 0. \tag{30}$$

---

[4]The quantities $T^\mu{}_\nu$ and $J^\mu$ transform as tensors under general coordinate transformations. Therefore, if we perform a coordinate transformation from the LAB system to any other (primed) system of the form $x^\mu = x^\mu(x')$, the energy-momentum tensor and $U(1)$ current in the new (primed) system take the form

$$T'^\mu{}_\nu = \frac{\partial x'^\mu}{\partial x^\rho}\frac{\partial x^\sigma}{\partial x'^\nu}T^\rho{}_\sigma, \qquad J'^\mu = \frac{\partial x'^\mu}{\partial x^\rho}J^\rho.$$

[5]This agrees with the holographic perfect fluid [5] in which there is no $U(1)$. To see this one needs to identify $\tilde{\mathscr{E}} = \mathscr{E}_{\text{Ref. [5]}} - \frac{1}{2}\rho V^2$, where $V$ is what is called $v$ in the present paper.

Using the thermodynamic relations (10) as well as the conservation equation

$$\partial_\mu J^\mu = \partial_0 n + \partial_i \left( n v^i \right) = 0 \,, \tag{31}$$

we obtain the following conserved entropy current

$$\partial_0 s + \partial_i \left( s v^i \right) = 0 \,. \tag{32}$$

Perfect fluids have therefore no entropy production and are thus non-dissipative.

From (29), we can derive a generalized Euler equation

$$\partial_0 \vec{v} + (\vec{v} \cdot \vec{\nabla})\vec{v} = -\frac{1}{\rho}\vec{\nabla}P - \frac{\vec{v}}{\rho}\left[ \partial_0 \rho + \partial_i(\rho v^i) \right] \,. \tag{33}$$

When $\rho$ is the mass density in a Galilei-fluid, $\rho_G = mn$, the last term vanishes and the equation reduces to the well-known (sourceless) Euler equation by means of (31). In the absence of boost invariance, one needs to have more information about the kinetic mass density to further determine the correction to the Euler equation. For perturbations away from a Galilei fluid that break boost symmetry, we can parametrize $\rho = \rho_G + \delta\rho$, we find deviations to the Euler equation,

$$\partial_0 \vec{v} + (\vec{v} \cdot \vec{\nabla})\vec{v} + \frac{1}{\rho_G}\vec{\nabla}P = \frac{\delta\rho}{\rho_G^2}\vec{\nabla}P - \frac{\vec{v}}{\rho_G}\left[ \partial_0(\delta\rho) + \partial_i(\delta\rho\, v^i) \right] \,. \tag{34}$$

Similarly for the relativistic case, which we discuss in the next section.

The Navier-Stokes equation is an extension of the Euler equation to include viscosity and diffusion. This is beyond the level of the perfect fluid description and will be discussed in [3].

Although this paper focuses on symmetries of the underlying theory of which we are making a fluid approximation, that show up in Ward identities and transformation properties of the currents, it is interesting to study symmetries of the perfect fluid equations of motion. For this we refer to [6] and references therein.

## 2.4 Boost invariant equations of state

We have $d+4$ variables $\mathscr{E}$, $P$, $\rho$, $n$ and $v^i$ and only $d+2$ conservation equations $\partial_\mu T^\mu{}_\nu = 0$ and $\partial_\mu J^\mu = 0$. In order to have a solvable system we need one equation of state $P = P(T, \mu, v^2)$ from which we can obtain $\rho$ and $n$ via (12) and $\mathscr{E}$ via the usual Euler relation. The expression for $\rho$ either follows from boost invariance or needs to be supplied by hand in such a way that it obeys (12).

### 2.4.1 Lorentz boost invariance

Consider the fluid in the static coordinates (19) and (20). In these coordinates the current corresponding to the Lorentz transformations is $\mathscr{J}_{\mu\nu}^\rho = x_\mu T^\rho{}_\nu - x_\nu T^\rho{}_\mu$, and the conservation law $\partial_\rho \mathscr{J}_{\mu\nu}^\rho = 0$ leads to $T_{\mu\nu} = T_{\nu\mu}$, where indices are lowered and raised by the Minkowski metric $\eta_{\mu\nu} = \text{diag}(-1, 1, ..., 1)$, and we set the speed of light $c = 1$. In particular this leads to $T^i{}_0 + T^0{}_i = 0$, from which it follows that

$$\mathscr{P}_i = \rho v^i = (\mathscr{E} + P)v^i \,. \tag{35}$$

The thermodynamic relation (10) becomes

$$(\mathscr{E} + P)(1 - v^2) = Ts + \mu n \,, \tag{36}$$

so that it is natural to define a new energy $\tilde{\mathscr{E}}$ (internal energy) such that

$$(\mathscr{E} + P)(1 - v^2) = Ts + \mu n = \tilde{\mathscr{E}} + P \,, \tag{37}$$

consistent with its previous definition $\tilde{\mathscr{E}} = \mathscr{E} - v^i \mathscr{P}_i$. From equations (35) and (37) it follows that $\rho$ is given by

$$\rho = \mathscr{E} + P = \frac{\tilde{\mathscr{E}} + P}{1 - v^2} \,. \tag{38}$$

Using the first equation in (29), the generalized Euler equation (33) applied to this case then becomes that of the standard relativistic form,

$$\partial_0 \vec{v} + (\vec{v} \cdot \vec{\nabla}) \vec{v} = -\frac{1 - v^2}{\tilde{\mathscr{E}} + P} \left( \vec{\nabla} P + \vec{v} \, \partial_0 P \right) . \tag{39}$$

We thus see that the expression for $\rho$ contains information about the properties of the system under boost transformations. Writing out the first law (24) in terms of $\tilde{\mathscr{E}}$, we obtain

$$\mathrm{d}\tilde{\mathscr{E}} = T \mathrm{d}s + \mu \mathrm{d}n - \frac{1}{2} \frac{\tilde{\mathscr{E}} + P}{1 - v^2} \mathrm{d}v^2 \,. \tag{40}$$

We can remove the term with $\mathrm{d}v^2$ by redefining $T, s, \mu$ and $n$ as follows

$$T = (1 - v^2)^{1/2} \tilde{T}, \qquad s = \frac{\tilde{s}}{(1 - v^2)^{1/2}}, \tag{41}$$

$$\mu = (1 - v^2)^{1/2} \tilde{\mu}, \qquad n = \frac{\tilde{n}}{(1 - v^2)^{1/2}} \,. \tag{42}$$

We then find

$$\tilde{\mathscr{E}} + P = \tilde{T} \tilde{s} + \tilde{\mu} \tilde{n}, \qquad \mathrm{d}\tilde{\mathscr{E}} = \tilde{T} \mathrm{d}\tilde{s} + \tilde{\mu} \mathrm{d}\tilde{n} \,. \tag{43}$$

This shows that $\tilde{\mathscr{E}}(\tilde{s}, \tilde{n})$ is independent of $v^2$, and therefore so is the pressure $P(\tilde{T}, \tilde{\mu})$, as also follows from

$$\mathrm{d}P = \tilde{s} \mathrm{d}\tilde{T} + \tilde{n} \mathrm{d}\tilde{\mu} \,. \tag{44}$$

Of course this independence is a consequence of boost invariance.

Using the expression for $\rho$ in the LAB frame we obtain

$$T^0{}_0 = -(\tilde{\mathscr{E}} + P) \frac{1}{1 - v^2} + P, \tag{45}$$

$$T^i{}_0 = -\left(\tilde{\mathscr{E}} + P\right) \frac{v^i}{1 - v^2}, \tag{46}$$

$$T^i{}_j = \left(\tilde{\mathscr{E}} + P\right) \frac{v^i v^j}{1 - v^2} + P \delta^i{}_j, \tag{47}$$

$$J^0 = \frac{\tilde{n}}{(1 - v^2)^{1/2}}, \tag{48}$$

$$J^i = \frac{\tilde{n}}{(1 - v^2)^{1/2}} v^i \,. \tag{49}$$

This is the standard form of a relativistic energy-momentum tensor and $U(1)$ charge current of a perfect fluid. If we reinstate factors of $c$ the energy-momentum tensor and $U(1)$ current of a relativistic perfect fluid take the following form

$$T^0{}_0 = -(\tilde{\mathscr{E}} + P) \frac{1}{1 - \frac{u^2}{c^2}} + P, \qquad T^i{}_0 = -(\tilde{\mathscr{E}} + P) \frac{u^i}{1 - \frac{u^2}{c^2}}, \tag{50}$$

$$T^0{}_i = \frac{\tilde{\mathscr{E}} + P}{c^2} \frac{u^i}{1 - \frac{u^2}{c^2}}, \qquad T^i{}_j = \frac{\tilde{\mathscr{E}} + P}{c^2} \frac{u^i u^j}{1 - \frac{u^2}{c^2}} + P \delta^i{}_j, \tag{51}$$

$$J^0 = \frac{\tilde{n}}{\left(1 - \frac{u^2}{c^2}\right)^{1/2}}, \qquad J^i = \frac{\tilde{n} u^i}{\left(1 - \frac{u^2}{c^2}\right)^{1/2}}, \tag{52}$$

where we denote the velocity by $u^i$. This can be written more compactly in terms of the covariant velocity $U^\mu = \frac{dx^\mu}{d\tau}$ where $x^\mu = (t, x^i)$ and where $\tau$ is the proper time $-c^2 dt^2 + d\vec{x}^2 = -c^2 d\tau^2$. Note that sometimes the 4-velocity is defined with $x^0 = ct$ but here we take $x^0 = t$. The components of $U^\mu$ are

$$U^0 = \frac{1}{\left(1 - \frac{u^2}{c^2}\right)^{1/2}}, \qquad U^i = \frac{u^i}{\left(1 - \frac{u^2}{c^2}\right)^{1/2}}, \tag{53}$$

$$U_0 = \frac{-c^2}{\left(1 - \frac{u^2}{c^2}\right)^{1/2}}, \qquad U_i = \frac{u^i}{\left(1 - \frac{u^2}{c^2}\right)^{1/2}}, \tag{54}$$

so that $U^\mu U_\mu = -c^2$. The covariant expressions for the energy-momentum tensor and $U(1)$ current then reduce to the well-known expressions

$$T^\mu{}_\nu = \frac{\tilde{\mathscr{E}} + P}{c^2} U^\mu U_\nu + P\delta^\mu_\nu, \qquad J^\mu = \tilde{n} U^\mu. \tag{55}$$

The other boost invariant cases can be obtained as limits of the relativistic case. We will discuss these cases next.

### 2.4.2 Carroll boost invariance

The generator of Lorentz boosts (with $c$ included) reads $L_i = \frac{1}{c} x^i \partial_0 + ct\partial_i$. This generator acts on the coordinates and induces the infinitesimal transformation $\delta x^\mu = \epsilon^i L_i x^\mu$. Sending $c$ to zero (the Carrollian limit) we obtain the generator $C_i = x^i \partial_0 \equiv C_i^\mu \partial_\mu$ which generates Carroll boost transformations $t' = t + \bar{v}^i x^i$ and $x'^i = x^i$ where $\bar{v}^i$ is a constant vector with dimensions of an inverse velocity. When there is Carroll boost invariance we have the additional conserved current

$$\partial_\mu \left(T^\mu{}_\nu C_i^\nu\right) = \partial_\mu \left(T^\mu{}_0 x^i\right) = T^i{}_0 = 0. \tag{56}$$

Equation (19) tells us that in this case $\mathscr{E} = -P$. From (10) it follows that $-v^i \mathscr{P}_i = Ts + \mu n$. Hence we have $-v^i \mathscr{P}_i = \tilde{\mathscr{E}} + P$ so that $Ts + \mu n = \tilde{\mathscr{E}} + P$. We thus obtain an equation of state for which

$$\rho = -\frac{\tilde{\mathscr{E}} + P}{v^2}. \tag{57}$$

The first law (24) in terms of $\tilde{\mathscr{E}}$ becomes

$$\mathrm{d}\tilde{\mathscr{E}} = T\mathrm{d}s + \mu\mathrm{d}n + \frac{1}{2}\frac{\tilde{\mathscr{E}} + P}{v^2}\mathrm{d}v^2. \tag{58}$$

We can remove the $\mathrm{d}v^2$ term by defining

$$T = \sqrt{v^2}\tilde{T}, \qquad s = \frac{1}{\sqrt{v^2}}\tilde{s}, \qquad \mu = \sqrt{v^2}\tilde{\mu}, \qquad n = \frac{1}{\sqrt{v^2}}\tilde{n}, \tag{59}$$

so that

$$\mathrm{d}\tilde{\mathscr{E}} = \tilde{T}\mathrm{d}\tilde{s} + \tilde{\mu}\mathrm{d}\tilde{n}, \qquad \tilde{\mathscr{E}} + P = \tilde{T}\tilde{s} + \tilde{\mu}\tilde{n}. \tag{60}$$

We thus see that the thermodynamics is indepenent of $v$ due to the presence of a boost symmetry.

In the LAB frame (19) and (20) we find

$$T^0{}_0 = P, \tag{61}$$

$$T^i{}_0 = 0, \tag{62}$$

$$T^0{}_j = -(\tilde{\mathscr{E}} + P)\frac{v^j}{v^2}, \tag{63}$$

$$T^i{}_j = P\delta^i_j - (\tilde{\mathscr{E}} + P)\frac{v^i v^j}{v^2}, \tag{64}$$

$$J^0 = \frac{\tilde{n}}{\sqrt{v^2}}, \tag{65}$$

$$J^i = \frac{\tilde{n}}{\sqrt{v^2}}v^i. \tag{66}$$

It is interesting that some non-trivial fluid description survives in this limit, but it would be nice to have some more concrete and intuitive examples of these Carrollian fluids.

### 2.4.3 Massless Galilei boost invariance

Sending $c$ to infinity in the Lorentz generator $-iL_i = \frac{1}{c}x^i\partial_0 + ct\partial_i$ we obtain the generator of Galilean transformations $G_i = t\partial_i = G^\mu_i\partial_\mu$. In the case of a massless Galilean theory the boost Ward identity is

$$\partial_\mu\left(T^\mu{}_\nu G^\nu_i\right) = \partial_\mu\left(T^\mu{}_i t\right) = T^0{}_i = 0. \tag{67}$$

This means that the momentum vanishes and thus we have an equation of state for which

$$\rho = 0. \tag{68}$$

In this case the thermodynamic relations (10) become

$$\mathscr{E} = Ts + \mu n - P, \qquad d\mathscr{E} = Tds + \mu dn, \tag{69}$$

and there is no need to redefine the energy and the other thermodynamic variables. In the LAB frame the form of the energy-momentum tensor reads

$$T^0{}_0 = -\mathscr{E}, \tag{70}$$

$$T^0{}_i = 0, \tag{71}$$

$$T^i{}_0 = -(\mathscr{E} + P)v^i, \tag{72}$$

$$T^i{}_j = P\delta^i_j, \tag{73}$$

$$J^0 = n, \tag{74}$$

$$J^i = nv^i. \tag{75}$$

This can be obtained by sending $c$ to infinity in (50)–(52).

The equations of motion $\partial_\mu T^\mu{}_\nu = 0$ and $\partial_\mu J^\mu = 0$ do not involve any time derivatives of the velocity $v^i$ and so we will not consider this case further.

### 2.4.4 Massive Galilei boost invariance

In the case of a Bargmann (massive Galilean) invariant theory we have the boost Ward identity $T^0{}_i = mJ^i$ [7–12] so that

$$\rho = mn, \tag{76}$$

where $m$ is a mass parameter of the theory. Hence $\rho$ is really a mass density, the mass times the particle number density. Now the thermodynamic relations (24) become

$$\tilde{\mathscr{E}} = Ts - P + \mu mn, \qquad d\tilde{\mathscr{E}} = Tds - \frac{1}{2}mndv^2 + \mu mdn. \tag{77}$$

We can remove the $dv^2$ terms by defining the variables

$$\hat{\mathscr{E}} = \tilde{\mathscr{E}} + \frac{1}{2}mnv^2 = \mathscr{E} - \frac{1}{2}mnv^2, \qquad \hat{\mu} = \mu + \frac{1}{2}mv^2, \tag{78}$$

so that

$$\hat{\mathscr{E}} = Ts - P + \hat{\mu}n, \qquad d\hat{\mathscr{E}} = Tds + \hat{\mu}dn. \tag{79}$$

Here $\hat{\mathscr{E}}$ is the internal energy and $\frac{1}{2}mnv^2$ is the kinetic energy of a fluid element with mass density $mn$ and velocity $v^i$.

The energy-momentum tensor and $U(1)$ current in the LAB frame read

$$T^0{}_0 = -\left(\hat{\mathscr{E}} + \frac{1}{2}mnv^2\right), \tag{80}$$

$$T^i{}_0 = -\left(\hat{\mathscr{E}} + P + \frac{1}{2}mnv^2\right)v^i, \tag{81}$$

$$T^0{}_i = mnv^i, \tag{82}$$

$$T^i{}_j = P\delta^i_j + mnv^iv_j, \tag{83}$$

$$J^0 = n, \tag{84}$$

$$J^i = nv^i. \tag{85}$$

The corresponding equations of motion agree with the standard expressions for a non-relativistic fluid such as the Euler equation of motion and the equation of continuity. We see that the Ward identity $\rho = mn$ makes mass a conserved quantity. The $U(1)$ current $J^\mu$ in this case is interpreted as the mass current of the theory.

In all cases (Lorentz, Carroll, Galilei and Bargmann) discussed so far there are additional currents resulting from boost invariance from which we can determine $\rho$. In the case with no boost symmetries we thus need to supply by hand an expression for $\rho$ in terms of the other fluid variables that is consistent with (12).

## 2.5 Scale invariance

We can add scale symmetries in which case the equation of state for $P$ follows from the existence of a dilatation current $\partial_\mu(T^\mu{}_\nu D^\nu) = 0$, where $-iD = zt\partial_t + x^i\partial_i$ generates dilatations $t \to \lambda^z t$ and $x^i \to \lambda x^i$ with dynamical exponent $z$. The Ward identity associated with scale symmetry is thus

$$\partial_0\left(ztT^0{}_0 + x^jT^0{}_j\right) + \partial_i\left(ztT^i{}_0 + x^jT^i{}_j\right) = zT^0{}_0 + T^i{}_i = 0, \tag{86}$$

which implies an equation of state for the pressure $P$:

$$dP = z\mathscr{E} - v^i\mathscr{P}_i = z\mathscr{E} - \rho v^2 = z\tilde{\mathscr{E}} + (z-1)\rho v^2. \tag{87}$$

This equation still holds in the absence of boosts, and the total symmetry algebra is the Lifshitz algebra consisting of time and spatial translations, rotations and Lifshitz dilatations. If we also add the boosts, it must be that the dilatation current $D$ forms a closed algebra with the other currents $H$, $P_i$, $J_{ij}$ and the generator for boosts. For Galilean, Bargmann or Carrollian boosts

this algebra exists for any value of $z$. However in the case of Lorentz boosts this only works for $z = 1$.

Let us consider the Lorentz invariant case in which $\rho = \frac{\tilde{\mathscr{E}}+P}{1-v^2}$ with both $P$ and $\tilde{\mathscr{E}}$ independent of $v^2$ as discussed in the previous subsection. This is only compatible with scale invariance if we set $z = 1$ in agreement with the known fact that we can only add scale symmetries to the Poincaré algebra for $z = 1$. If we consider the Galilean case with $\rho = 0$ there is no restriction on what $z$ should be. For Carroll boost symmetries we have $\rho = -(\tilde{\mathscr{E}} + P)/v^2$ with $\tilde{\mathscr{E}}$ and $P$ independent of $v^2$. The equation of state (87) gives $(d + z - 1)P = \tilde{\mathscr{E}}$ which is independent of $v^2$ for any $z$. Hence we can have scale symmetries with any value of $z$ for Carroll boost invariant perfect fluids. This is consistent with the fact that we can add dilatations to the Carroll algebra with any $z$. For the Bargmann case, something special happens, and we discuss this in a separate subsection next.

## 2.6 No-go theorem for perfect Schrödinger fluids

Finally, let us consider the Bargmann case. In this case we have $\rho = mn \neq 0$ with $\hat{\mathscr{E}} = \tilde{\mathscr{E}} + \frac{1}{2}\rho v^2$ the internal energy that together with $P$ is $v^2$ independent. From (87) we learn that

$$dP = z\hat{\mathscr{E}} + \frac{z-2}{2}\rho v^2 . \tag{88}$$

From (79) it follows that $P$ is a function only of $T$ and $\hat{\mu}$. Since $s = \left(\frac{\partial P}{\partial T}\right)_{\hat{\mu}}$ and $n = \left(\frac{\partial P}{\partial \hat{\mu}}\right)_T$ and since $\hat{\mathscr{E}} = \hat{\mathscr{E}}(s, n)$ it follows that the combination $(dP - z\hat{\mathscr{E}})/n$ is a function of $T$ and $\hat{\mu}$ and not of $v^2$. We conclude that this is compatible with scale symmetries only for $z = 2$. On an algebraic level, in the case of the Bargmann algebra, we can add scale symmetries with general $z$ leading to the Schrödinger algebra with general $z$. Here we see that we cannot form a perfect Schrödinger fluid with $z \neq 2$ that fulfils our basic assumptions of having $H$, $P_i$ and $J_{ij}$ symmetries. Hence we have derived a no-go theorem.

We now give some further arguments as what the physical origin of this no-go theorem is. In the notation of e.g. [13], the Schrödinger algebra with $z \neq 2$ has the following non-vanishing commutation relations

$$
\begin{aligned}
[D, H] &= -izH , & [D, G_a] &= i(z-1)G_a , \\
[H, G_a] &= iP_a , & [D, P_a] &= -iP_a , \\
[D, N] &= i(z-2)N , & [P_a, G_b] &= i\delta_{ab}N ,
\end{aligned}
\tag{89}
$$

where $D, G_a, P_a, N, H$ stand for dilatations, Galilei-boosts, translations, number operator, and Hamiltonian. There are also rotation generators that simply rotate all generators with a vector index and commute with $D, N, H$.

In a system with scale invariance, there is a danger that the spectrum is continuous and therefore the partition function is not well-defined. This does not happen in conventional systems, because once we put the system in finite volume the spectrum becomes discrete which makes the partition function well-behaved. To make the spectrum discrete it is important that there is a relation between energy and momenta. Similarly, in this case we expect to need two relations which make the spectrum of both $H$ and $N$ discrete in finite volume in order to have well-defined thermodynamics.

Let's try to find operators $A$ in the theory which commute with all generators except $D$ and have a well-defined scaling dimension under $D$. If the scaling dimension of $A$ is non-zero, we can consistently impose $A = 0$. If the scaling dimension is zero, we can consistently impose $A = c\mathbf{1}$ for any constant $c$. As said above, without such restrictions, it seems hard to construct a representation where the spectrum of both $H$ and $N$ is not continuous.

The most general rotationally invariant operators is a combination of $H$, $N$, $P^2$, $G^2$, and $P \cdot G$. One can easily check that such an operator can only commute with $H$ and $P_a$ if it does not depend on $G^2$ and $P \cdot G$. In order for the operator to commute with $G_a$, it must be built from $N$ and $2NH - P^2$ having weights $z - 2$ and $-2$ under dilatations respectively.

We therefore see that for $z = 2$ we can impose $N = $ const and $2NH - P^2 = 0$. If the spectrum of $P$ is discrete, the spectrum of both $N$ and $H$ will also be discrete and the partition functions may be well-defined.

For $z \neq 2$, if we wish to impose two constraints, we can only impose $N = 2NH - P^2 = 0$, but then $P^2 = 0$ and the representation is pathological.

If we wish to impose only one constraint, we can impose $N = 0$, but then we are back in the Galilean case. We can also impose $2NH - P^2 = 0$, but then only the spectrum of $NH$ is constrained. Finally, we can also impose (we could call this combination the Casimir of the Schrödinger algebra for $z \neq 2$)

$$N^{\frac{2}{z-2}}(2NH - P^2) = \text{const}, \tag{90}$$

but again this does not constrain both $N$ and $H$.

One can for example get (90) if we start with a particle in one-dimension higher and interpret $N$ and $H$ as the two light-cone components of this higher dimensional momenta. Then (90) can be rewritten as

$$(n_\rho p^\rho)^{\frac{2}{z-2}}(\eta_{\mu\nu} p^\mu p^\nu) = \text{const}, \tag{91}$$

with $n_\mu$ a null vector and we can build a unitary representation of the Schrödinger algebra with $z \neq 2$ on a single-particle Hilbert space in one dimension higher. But clearly, if we put the system in a finite volume in one dimension lower, then the spectrum of energies remains continuous because the momentum in the additional dimension remains continuous. We could choose to discretize this additional momentum, but then the thermodynamics will probably no longer be extensive in the original volume once we take the continuum limit.

Interestingly, the Schrödinger algebra with $z \neq 2$ also appears in [14] where it has the name $DISIM_b(2)$. A single particle action with this symmetry is given in equation (17) in that paper[6], and the dispersion relation in (18) which has exactly the form (91).

The conclusion seems to be that while unitary representations of the Schrödinger algebra with $z \neq 2$ exist with energies bounded from below, these representations typically have continuous spectrum and therefore there is no well-defined partition function in finite volume which respects the full Schrödinger symmetry[7].

One might suggest that black branes in Schrödinger space-times with $z \neq 2$ and their holographically dual finite temperature field theories are a counterexample to our no-go theorem. However on closer inspection one sees that all cases of such black branes violate one of our assumptions. Either the dimensionality of the problem has changed because the Schrödinger part of the metric depends explicitly on one of the internal coordinates as in [15, 16], or because the metric is not rotationally symmetric like the one in [17], or because the geometry

---

[6]The deformation parameter $b$ in [14] is related to the dynamical exponent via $b = (1-z)^{-1}$, and their dilatation operator is called $N$ and is related to $D$ by $(1-z)N = D$.

[7]It is possible to write down field theories with both Galilean boost and dilatation symmetries for general $z$. An example is

$$\mathscr{L} = -\varphi^\alpha \left( \partial_t \theta + \frac{1}{2} \partial_i \theta \partial_i \theta \right) - \frac{1}{2} \partial_i \varphi \partial_i \varphi, \qquad \alpha = 2 \frac{d-z+2}{d+z-2}. \tag{92}$$

The scaling dimensions of the two real scalars are $[\varphi] = (d + z - 2)/2$ and $[\theta] = z - 2$. This model has full Schrödinger symmetries for general $z$. For $z = 2$, it becomes the usual Schrödinger model with $\phi = \frac{1}{\sqrt{2}} \varphi e^{i\theta}$ the Schrödinger wavefunction. For $z \neq 2$ the quantisation of the model seems more problematic. It would be interesting to study this further.

is only conformally Schrödinger in Einstein frame [18] so that there is hyperscaling violation and thus no strict scale invariance. This latter case also includes the work [19].

We mention that in our derivation of the above no-go results we assumed that $\rho \neq 0$. This is physically motivated because we wish to consider systems with nonzero momentum. However, as we discussed at the end of the previous subsection, there are no restrictions on $z$ if we consider systems with massless ($\rho = 0$) Galilean symmetries. For a concrete example see [20].

## 2.7 Hyperscaling violation and charge anomalous dimension

We can generalize our no-go theorem by going away from scale invariant fluids to local thermodynamical systems that have a notion of scale covariance where the scale symmetry is broken due to the presence of a nonzero hyperscaling violation exponent $\theta$ or an anomalous dimension $\alpha$ for the charge current. Hyperscaling violation means that in all the scalings of the extensive thermodynamic quantities we replace the dimension of space $d$ by $d - \theta$ where $\theta$ is the hyperscaling violation exponent. This means that the entropy density would scale as $\delta s = (d - \theta)\lambda s$ when we scale space and time as $\delta x^i = -\lambda x^i$ and $\delta t = -z\lambda t$. A nonzero $\theta$ can e.g. be observed in critical systems above their upper critical dimension or e.g. when there are gapless fermionic excitations above a $d - 1$ dimensional Fermi surface. It effectively changes the thermodynamic dimensionality of the system. The value of $\theta$ can be negative and positive. Another scaling relation that can be modified is that of the charge density $n$ by including an anomalous scaling dimension. If we consider a system with vanishing $\theta$ the standard scaling of the charge density is simply $\delta n = d\lambda n$. When we include an anomalous scaling $\alpha$ this is modified to $\delta n = (d - \alpha)\lambda n$. See e.g. [21] for field theory realizations of this and [22] for a proposal of its relevance for cuprate superconductors. The parameters $\theta$ and $\alpha$ were found in holography in [23, 24] and [25, 26], respectively.

When we turn on both of these scaling parameters we have the following scalings of the intensive variables $T$, $\mu$, $v^2$ and the pressure $P$,

$$\delta T = z\lambda T\,, \quad \delta\mu = (z + \alpha)\lambda\mu\,, \quad \delta v^2 = 2(z-1)\lambda v^2\,, \quad \delta P = (d - \theta + z)\lambda P\,. \tag{93}$$

It then follows from the Gibbs–Duhem relation (12) as well as the Euler relation (the first equation of (24)) that the equation of state is

$$(d - \theta)P = z\tilde{\mathscr{E}} + \alpha\mu n + (z-1)\rho v^2\,. \tag{94}$$

Let us next see when this can be compatible with Galilean boost symmetry for which $\rho = n$. Here we set the mass parameter $m$ relating $\rho$ and $n$ in the Bargmann case equal to one. Writing the equation of state in terms of $\hat{\mathscr{E}}$ and $\hat{\mu}$, defined in (78), we obtain

$$(d - \theta)P = z\hat{\mathscr{E}} + \alpha\hat{\mu}n + \frac{z - 2 - \alpha}{2}nv^2\,. \tag{95}$$

Since $P$, $\hat{\mathscr{E}}$, $\hat{\mu}$ and $n$ are independent of $v^2$ for a Galilean invariant system we find that for $n \neq 0$ the equation of state (95) is only consistent for $\alpha = z - 2$. For this specific value of $\alpha$ the charge density $n$ scales as $\delta n = (d + 2 - z)\lambda n$ so that the total charge $N$ in a volume $V$ scales as $\delta N = (2 - z)\lambda N$. This is precisely the scaling dimension of the mass or particle number operator in the Schrödinger algebra for general $z$. Hence, for $\alpha = z - 2$ we can find a realization of the Schrödinger transformations for general $z$ on the fluid variables that leaves the equation of state invariant. We do however not have a Ward identity (conserved current) for dilatations due to the nonzero values of $\theta$ and $\alpha$.

Whereas systems with a suitable value of $\alpha$ are compatible with Galilean boost invariance there is no such statement for the hyperscaling violation exponent $\theta$. In other words if $\alpha \neq z-2$

there is no value of $\theta$ for which the system is compatible with Galilean boost symmetry and vice versa hyperscaling violating and Galilean boost symmetric thermodynamic systems require $\alpha = z - 2$. The black branes of [18] are of this type since they have $z \neq 1$, $\theta \neq 0$ and $\alpha = z - 2$.

It is also interesting to study what happens for $z = 1$ because this is a special case. For $z = 1$ we can write (94) in terms of the Lorentzian quantities $\tilde{\mu}$ and $\tilde{n}$, defined in (42), as $(d - \theta)P = \tilde{\mathscr{E}} + \alpha \tilde{\mu} \tilde{n}$ which consists entirely of $v^2$ independent quantities so that any value of $\theta$ and $\alpha$ is consistent with Lorentz boost symmetry.

Our no-go theorem of the previous subsection can thus be generalized to the following statement. If one observes scaling exponents $(z, \theta, \alpha)$ with $z \neq 1$ and $\alpha \neq z - 2$, in the presence of a nonzero $\rho$, the system cannot be Galilean boost invariant[8].

## 2.8 Geometry and Equilibrium Partition Function

The geometry on which fluids without boosts are defined is the geometry of Aristotelian or absolute spacetime where 'absolute' is meant in the sense of the existence of an absolute rest frame. This is described by the following metric objects: $\tau_\mu$ and $h_{\mu\nu}$ that do not transform under any kind of local tangent space transformation. The signature of $h_{\mu\nu}$ is $(0, 1, \ldots, 1)$. This is different from (torsional) Newton–Cartan geometry in which $h_{\mu\nu}$ transforms under local Galilean boosts. In Carrollian geometry it is $\tau_\mu$ but not $h_{\mu\nu}$ that transforms under local Carrollian boosts. In Lorentzian geometry both $\tau_\mu$ and $h_{\mu\nu}$ transform under local Lorentz transformations in such a way that $\gamma_{\mu\nu} = -\tau_\mu \tau_\nu + h_{\mu\nu}$ remains invariant. One could say that torsional Newton–Cartan, Carrollian and Lorentzian geometries are all special cases of the geometry of absolute spacetime in which $\tau_\mu$ and $h_{\mu\nu}$ are assigned specific local tangent space transformations.

Because of the signature of $h_{\mu\nu}$ we can decompose it into vielbeins $h_{\mu\nu} = \delta_{ab} e_\mu^a e_\nu^b$, where $a = 1, \ldots, d$ and $\mu$ takes $d + 1$ values. The spatial vielbeins $e_\mu^a$ transform under local $SO(d)$ transformations. The square matrix $(\tau_\mu, e_\mu^a)$ is invertible and its inverse will be denoted by $(-v^\mu, e_a^\mu)$ where we have

$$v^\mu \tau_\mu = -1 , \qquad v^\mu e_\mu^a = 0 , \qquad e_a^\mu \tau_\mu = 0 , \qquad e_a^\mu e_\mu^b = \delta_a^b . \tag{96}$$

The completeness relation is $-v^\mu \tau_\nu + e_a^\mu e_\nu^a = \delta_\nu^\mu$. The determinant of $(\tau_\mu, e_\mu^a)$ will be denoted by $e$, i.e. $e = \det(\tau_\mu, e_\mu^a)$, and we also define $h^{\mu\nu} \equiv e_a^\mu \delta^{ab} e_b^\nu$ which satisfies $h^{\mu\rho} h_{\rho\nu} = \delta_\nu^\mu + v^\mu \tau_\nu$. The general framework of this absolute spacetime geometry can be derived from [27, 28].

Let us assume that the background described by $\tau_\mu$ and $h_{\mu\nu}$ has a time-translation symmetry (but is otherwise arbitrary) generated by the vector $\beta^\mu$ which satisfies the Killing equations

$$\mathscr{L}_\beta \tau_\mu = 0 , \qquad \mathscr{L}_\beta h_{\mu\nu} = 0 . \tag{97}$$

The vector $\beta^\mu$ leads to a preferred choice of local temperature and velocity defined by

$$T \equiv 1/(\tau_\mu \beta^\mu) , \qquad u^\mu \equiv T \beta^\mu , \tag{98}$$

where the velocity $u^\mu$ satisfies $u^\mu \tau_\mu = 1$.

In order to construct the hydrostatic partition function [29, 30] we write the most general expansion in derivatives of the background fields $\tau_\mu$ and $h_{\mu\nu}$ where we assume that the Killing equations (97) are obeyed and where we treat $\beta^\mu$ as a fixed vector that is not varied when computing the variation of the partition function. At zeroth order in derivatives there are two scalars that one can be build. These are

$$T , \quad u^2 , \tag{99}$$

---

[8]As an application of this result we conclude that the relations suggested in [22] for which $z = 4/3$, $\theta = 0$ and $\alpha = 2/3$ (called $-\Phi$ in [22]) are incompatible with a boost symmetry.

where we defined $u^2 = h_{\nu\rho}u^\nu u^\rho$.

Let us consider the hydrostatic partition function up to zeroth order in derivatives[9], i.e.

$$S = \int d^{d+1}x\, e\, P(T, u^2). \tag{100}$$

Restricting to zeroth order is sufficient at the level of perfect fluids. Going beyond this will be discussed in [3, 4]. Since we vary the background sources keeping $\beta^\mu$ fixed we have $\delta T = -Tu^\mu \delta\tau_\mu$ and $\delta u^\mu = -u^\mu u^\rho \delta\tau_\rho$. Using further that $\delta e = e\left(-\nu^\mu \delta\tau_\mu + \frac{1}{2}h^{\mu\nu}\delta h_{\mu\nu}\right)$. We define the responses $T^\mu$ and $T^{\mu\nu}$ as follows

$$\delta S = \int d^{d+1}x\, e\left(-T^\mu \delta\tau_\mu + \frac{1}{2}T^{\mu\nu}\delta h_{\mu\nu}\right). \tag{101}$$

We then find

$$T^\mu = P\nu^\mu + \left(\frac{\partial P}{\partial T}\right)_{u^2} Tu^\mu + 2\left(\frac{\partial P}{\partial u^2}\right)_T u^2 u^\mu, \tag{102}$$

$$T^{\mu\nu} = Ph^{\mu\nu} + 2\left(\frac{\partial P}{\partial u^2}\right)_T u^\mu u^\nu. \tag{103}$$

We now take $P$ to be the pressure and using the thermodynamic relations $\left(\frac{\partial P}{\partial T}\right)_{u^2} = s$, $\left(\frac{\partial P}{\partial u^2}\right)_T = \frac{1}{2}\rho$, $sT + \rho u^2 = \mathcal{E} + P$ as well as $u^\mu = -\nu^\mu + h^{\mu\rho}h_{\rho\nu}u^\nu$, which follows from the completeness relation and $\tau_\mu u^\mu = 1$. Then we obtain

$$T^\mu = \mathcal{E}u^\mu + Ph^{\mu\rho}h_{\rho\nu}u^\nu, \tag{104}$$

$$T^{\mu\nu} = Ph^{\mu\nu} + \rho u^\mu u^\nu. \tag{105}$$

The energy-momentum tensor $T^\mu{}_\nu$ is given by

$$T^\mu{}_\nu = -T^\mu \tau_\nu + T^{\mu\rho}h_{\rho\nu}, \tag{106}$$

which can further be written as

$$T^\mu{}_\nu = -(\mathcal{E} + P)u^\mu \tau_\nu + P\delta^\mu_\nu + \rho u^\mu u^\rho h_{\rho\nu}. \tag{107}$$

This is the spacetime covariant form of our new perfect fluid energy-momentum tensor. In the LAB frame with $\nu^\mu = -\delta^\mu_0$, $\tau_\mu = \delta^0_\mu$, $h_{0\mu} = 0$, $h^{0\mu} = 0$, $h_{ij} = \delta_{ij}$, $h^{ij} = \delta^{ij}$ and $u^i = v^i$ the expressions (104)–(106) become equal to (19).

## 3 Speed of sound

In this section, we derive new formulas for the speed of sound. As we will show, the standard "Landau-Lifshitz" formula for the sound speed no longer holds in the absence of boost symmetry, and needs to be generalized. In our theory, this is because the fluctuations of the perfect fluid energy-momentum tensor involves also the fluctuations of the kinetic mass density.

We first consider fluids at rest and derive a universal formula for the speed of sound without assuming a particular form for an equation of state. This means we expand the fluid velocity as $v^i = v^i_0 + \delta v^i$, $\mathcal{E} = \mathcal{E}_0 + \delta\mathcal{E}$, $P = P_0 + \delta P$, $\rho = \rho_0 + \delta\rho$ with $v^i_0 = 0$. The extension to boosted fluids with $v^i_0 \neq 0$ is given in the appendix. For the purpose of comparing with boosted fluids, we will write everything in terms of the internal energy $\tilde{\mathcal{E}}$, but notice that for fluids at rest

---

[9]We thank Kristan Jensen for useful discussions about the construction of the hydrostatic partition function.

we have $\tilde{\mathscr{E}}_0 = \mathscr{E}_0$, and $\delta\tilde{\mathscr{E}} = \delta\mathscr{E}$. The fluctuation equations that follow from (28) and current conservation are

$$
\begin{align}
0 &= \partial_0 \delta n + n_0 \partial_i \delta v^i, \tag{108}\\
0 &= \partial_0 \delta\tilde{\mathscr{E}} + \left(\tilde{\mathscr{E}}_0 + P_0\right)\partial_i \delta v^i, \tag{109}\\
0 &= \rho_0 \partial_0 \delta v^i + \partial_i \delta P. \tag{110}
\end{align}
$$

Notice that $\delta\rho$ does not appear here, because the fluid is at rest in equilibrium. We will write $P = P(\tilde{\mathscr{E}}, n, v^2)$ so that we have $\delta P = \left(\frac{\partial P_0}{\partial \tilde{\mathscr{E}}_0}\right)_{n_0} \delta\tilde{\mathscr{E}} + \left(\frac{\partial P_0}{\partial n_0}\right)_{\tilde{\mathscr{E}}_0} \delta n$ where we dropped the dependence on $v^2$ because at leading order $v_0^i = 0$ and $\delta v^2 = 0$ up to first order. Going to Fourier space, using $\delta\tilde{\mathscr{E}} = e^{-i\omega t + i k^i x^i} \delta\tilde{\mathscr{E}}(\omega, k)$ and similarly for $\delta v^i$ and $\delta n$, and defining $\delta v_{\parallel} = \frac{k^i}{k}\delta v^i$ we can derive the following equation for $\delta v_{\parallel}$

$$
\rho_0 \omega^2 \delta v_{\parallel} - k^2 \left[\left(\tilde{\mathscr{E}}_0 + P_0\right)\left(\frac{\partial P_0}{\partial \tilde{\mathscr{E}}_0}\right)_{n_0} + n_0 \left(\frac{\partial P_0}{\partial n_0}\right)_{\tilde{\mathscr{E}}_0}\right]\delta v_{\parallel} = 0, \tag{111}
$$

so that the speed of sound $v_s^2$ is

$$
v_s^2 = \frac{\tilde{\mathscr{E}}_0 + P_0}{\rho_0}\left(\frac{\partial P_0}{\partial \tilde{\mathscr{E}}_0}\right)_{n_0} + \frac{n_0}{\rho_0}\left(\frac{\partial P_0}{\partial n_0}\right)_{\tilde{\mathscr{E}}_0}. \tag{112}
$$

This can be rewritten as follows. From general thermodynamics with $P = P(\tilde{\mathscr{E}}, n, v^2)$ using the first law for $\delta\tilde{\mathscr{E}}$ as well as the fact that $\tilde{\mathscr{E}} + P = sT + \mu n$ we can show that

$$
\left(\frac{\partial P}{\partial n}\right)_{\frac{s}{n}, v^2} = \frac{\tilde{\mathscr{E}} + P}{n}\left(\frac{\partial P}{\partial \tilde{\mathscr{E}}}\right)_{n, v^2} + \left(\frac{\partial P}{\partial n}\right)_{\tilde{\mathscr{E}}, v^2}. \tag{113}
$$

Applying this to perturbations around a zero velocity thermodynamic configuration we find that the speed of sound (112) can be written as

$$
v_s^2 = \frac{n_0}{\rho_0}\left(\frac{\partial P_0}{\partial n_0}\right)_{\frac{s_0}{n_0}}. \tag{114}
$$

This generalizes the "Landau-Lifshitz" formula for the speed of sound, $v_s^2 = \left(\frac{\partial P_0}{\partial \rho_0}\right)_{\frac{s_0}{n_0}}$ that holds for the case when there is Galilei boost symmetry with $\rho = mn$.

When we are dealing with a scale invariant system, because $v_0^i = 0$, we can use that $dP_0 = z\tilde{\mathscr{E}}_0$. It follows that $\left(\frac{\partial P_0}{\partial n_0}\right)_{\tilde{\mathscr{E}}_0} = 0$ so that for a Lifshitz perfect fluid the speed of sound is given by

$$
v_s^2 = \frac{z}{d}\frac{\tilde{\mathscr{E}}_0 + P_0}{\rho_0}. \tag{115}
$$

If on the other hand we do not assume scale invariance and also remove the $U(1)$ current we find

$$
v_s^2 = \frac{\tilde{\mathscr{E}}_0 + P_0}{\rho_0}\left(\frac{\partial P_0}{\partial \tilde{\mathscr{E}}_0}\right), \tag{116}
$$

which reduces for the relativistic case to the well-known expression (with $c = 1$) $v_s^2 = \left(\frac{\partial P_0}{\partial \tilde{\mathscr{E}}_0}\right)$.

In the appendix, we extend our results to the case of a boosted fluid, and study the speed of sound propagating in a fluid that has some velocity $v_0^i$ in some inertial frame.

# 4 Ideal gas of Lifshitz particles

In this section we consider Boltzmann, Bose, and Fermi gases of identical Lifshitz particles moving in $d$ spatial dimensions. The one-particle Hamiltonian has the form

$$H_1 = \lambda \left( \vec{k}^2 \right)^{\frac{z}{2}} , \tag{117}$$

where $\lambda$ has the dimensions of $(\mathrm{kg})^{1-z}(\mathrm{m/s})^{2-z}$. We assume that $z \geq 1$. For $z = 1$ the Hamiltonian (117) describes a massless relativistic particle with $\lambda = c$. Another special case is when $z = 2$ and $\lambda = 1/(2m)$ in which case the system also includes Galilean boost symmetry. For arbitrary $z$, there is no boost symmetry, so it provides a concrete example illustrating the main ideas of our paper. Instead of boost symmetry, there is Lifshitz scale symmetry, manifested by the anisotropic (between time and space) scaling relations $t \to \alpha^z t, \vec{x} \to \alpha \vec{x}$. The dimensionful parameter $\alpha$ has no scaling weight, so that the Hamiltonian (117) obeys the Lifshitz scaling property of an energy, $H \to \alpha^{-z} H$. The Lagrangian of the Lifshitz particles takes the form

$$L = \lambda(z-1) \left( \vec{k}^2 \right)^{\frac{z}{2}} , \tag{118}$$

which for $z = 2$ reduces to the known result $L = \vec{k}^2/2m$. The relation between momentum and velocity is given by

$$\vec{k} = \left( \frac{1}{\lambda z} \right)^{\frac{1}{z-1}} \frac{1}{(\vec{v}^2)^{\frac{z-2}{2(z-1)}}} \vec{v} , \tag{119}$$

which reduces to $\vec{k} = m\vec{v}$ for $z = 2, \lambda = 1/(2m)$.

The Lagrangian scales in the same way as the Hamiltonian, such that the action $S = \int \mathrm{d}t \, L$ is indeed invariant under Lifshitz rescalings. The dynamics of free Lifshitz particles is of course very simple; they move on straight lines given by

$$\vec{x}(t) = \vec{v}_0 \, t + \vec{x}_0 , \qquad \vec{v}_0 = z E \frac{\vec{k}_0}{k_0^2} , \tag{120}$$

where the constants of motion, energy $E$ and momentum $\vec{k}_0$, are set by the initial conditions, and $E = \lambda \left( \vec{k}_0^2 \right)^{\frac{z}{2}}$ is the dispersion relation.

The Lifshitz scale symmetry strongly constrains the thermodynamics of the Lifshitz gas and the form of certain hydrodynamical quantities such as the speed of sound. Sound modes arise in the linear dispersion relation of the energy density fluctuations, with $k \equiv |\vec{k}|$,

$$\omega = v_s k . \tag{121}$$

This can be contrasted with the dispersion relation of a particle with Lifshitz scaling, i.e. $\omega = c_z k^z$, for some constant $c_z$ that has no scaling weight. The speed of sound $v_s$ therefore must have a scaling weight: as $\omega \to \alpha^{-z} \omega$ and $k \to \alpha^{-1} k$, we must have $v_s \to \alpha^{1-z} v_s$. Such a scaling behavior can only arise if $v_s$ has the right dependence on the temperature $T$ and on $\lambda$ to restore the correct SI units for a velocity. For instance, if there are no other scales in the problem other than $\lambda$ and $k_B T$ (e.g. in the absence of a chemical potential), we must have

$$v_s = \# (k_B T)^{\frac{z-1}{z}} \lambda^{\frac{1}{z}} , \tag{122}$$

for some numerical constant denoted by $\#$. In the next subsections, we compute this constant explicitly below for a Boltzmann, Bose, and Fermi gas of Lifshitz particles at finite density.

Some of the formulae presented in the remainder of this section can be found back in various places in the literature, or are rather straightforward generalizations of known results. We give references below. To the best of our knowledge, the formulae for the speed of sound are new however, as well as all velocity dependence of the partition function and kinetic mass density formula.

### 4.1 Boltzmann gas

The canonical partition function for a gas of $N$ non-interacting identical particles is given by

$$Z(N, T, V, \vec{v}) = \frac{1}{N!} \left[ Z_1(T, V, \vec{v}) \right]^N , \tag{123}$$

where $Z_1(T, V, \vec{v})$ is the single-particle partition function

$$Z_1(T, V, \vec{v}) = \frac{V}{h^d} \int d^d \vec{k} \, e^{-\beta H_1 + \beta \vec{v} \cdot \vec{k}} . \tag{124}$$

Here $\beta = \frac{1}{k_B T}$ is the inverse temperature, Planck's constant $h$ is written to make the partition function dimensionless, and we introduced a velocity vector as a chemical potential as discussed in Section 2. The volume is $V$ and the number of spatial dimensions is $d$. The result (123) is only valid when all particles occupy different states, so no two or more particles occupy the same energy level. This happens when the number of particles is much less than the number of thermally accessible energy levels in the system, which can be formulated as $N \ll Z_1$. This happens for sufficient low concentration and high temperature, where the gas behaves as a classical system. For a discussion, see e.g. [31].

At zero velocity $\vec{v} = 0$, we can explicitly evaluate

$$Z_1(T, V, \vec{v} = 0) = \frac{2V}{z} \left( \frac{\sqrt{\pi}}{h} \right)^d \frac{\Gamma\left[\frac{d}{z}\right]}{\Gamma\left[\frac{d}{2}\right]} \left( \frac{k_B T}{\lambda} \right)^{\frac{d}{z}} . \tag{125}$$

Notice that this partition function is scale invariant under $T \to \alpha^{-z} T$ combined with $V \to \alpha^d V$, which are the correct Lifhsitz scaling laws. The condition on low concentration can now be translated in terms of a thermal wavelength $\lambda_{th}$ as follows. The average interparticle spacing in the gas is set by the length scale $(V/N)^{1/d}$. Requiring $N \ll Z_1$ is then equivalent to [32]

$$\lambda_{th} \ll \left( \frac{V}{N} \right)^{\frac{1}{d}} , \qquad \lambda_{th}^{-d} \equiv \frac{Z_1}{V} = \frac{2}{z} \left( \frac{\sqrt{\pi}}{h} \right)^d \frac{\Gamma\left[\frac{d}{z}\right]}{\Gamma\left[\frac{d}{2}\right]} \left( \frac{k_B T}{\lambda} \right)^{\frac{d}{z}} . \tag{126}$$

When the thermal wavelength becomes of the same order as the interparticle distance, the gas can no longer be treated classically and one resorts to the Bose-description of the quantum Lifshitz gas, which we do in the next subsection.

The energy of the system follows from

$$U_0 \equiv \langle \tilde{E}_0 \rangle = -\frac{\partial \ln Z}{\partial \beta} = \frac{d}{z} N k_B T . \tag{127}$$

This is the equipartition theorem for Lifshitz particles, saying that the average kinetic energy is $(d/z)k_B T$ per particle. The subscript denotes that we are at zero velocity, $v = 0$. We extend this to non-zero velocities below. The dimensionless heat capacity formula then follows from

$$C_V = \left( \frac{\partial U}{\partial T} \right)_V , \qquad \hat{C}_V \equiv \frac{C_V}{N k_B} = \frac{d}{z} , \tag{128}$$

independent of the temperature. For $d = 3, z = 2$ this leads to the famous factor of 3/2 in the heat capacity of non-relativistic mono-atomic ideal gases. The generalization to arbitrary dimension just follows from counting degrees of freedom, and here we add also the extension to other values of $z$, which is a consequence of the simple scaling law in (125).

By taking differentials of the partition functions, we can derive all the thermodynamic quantities and rediscover the ideal gas law and energy densities $\tilde{\mathcal{E}}_0 = U_0/V = \langle \tilde{E}_0 \rangle / V$

$$PV = Nk_B T \,, \qquad \tilde{\mathcal{E}}_0 = \frac{d}{z} P_0 \,. \tag{129}$$

The last relation is in fact the $z$-trace Ward identity for the energy-momentum tensor. The Lifshitz weight for the pressure is then $-z - d$, i.e. it scales like $P \to \alpha^{-z-d} P$.

The entropy of the Lifshitz gas can be easily computed from $S = k_B(\beta U + \ln Z)$ and from this and the ideal gas law, we can compute the dimensionless heat capacity at constant pressure

$$C_P = T\left(\frac{\partial S}{\partial T}\right)_P \,, \qquad \hat{C}_P \equiv \frac{C_P}{Nk_B} = \frac{d}{z} + 1 \,, \tag{130}$$

leading to the adiabatic index

$$\gamma \equiv \frac{C_P}{C_V} = 1 + \frac{z}{d} \,. \tag{131}$$

The adiabatic index appears in the relation

$$PV^\gamma = constant \,, \tag{132}$$

which holds during adiabatic expansion or compression of the gas. The constant is a number determined from the initial volume and pressure. After the expansion or compression, the volume has changed with a fraction, and one then computes the pressure from (132). This equation is consistent with Lifshitz scaling, as $PV^\gamma$ has zero scaling weight. For large temperatures, the entropy behaves as

$$S \stackrel{T\to\infty}{=} k_B N \frac{d}{z} \ln T \,. \tag{133}$$

For non-zero velocities $\vec{v}$, the evaluation of the partition function in the canonical ensemble is less easy. For generic values of $z \geq 1$, one can integrate term by term in the power expansion of $v \equiv |\vec{v}|$, and the result is[10]

$$
\begin{aligned}
Z_1(T, V, \vec{v}) &= \frac{2V}{z}\left(\frac{\sqrt{\pi}}{h}\right)^d (\lambda\beta)^{-\frac{d}{z}} \sum_{n=0}^{\infty} \frac{\left(-\frac{\beta v}{2}\right)^{2n}}{n!} \frac{\Gamma\left[\frac{d+2n}{z}\right]}{\Gamma\left[\frac{d+2n}{2}\right]} (\lambda\beta)^{-\frac{2n}{z}} \\
&= Z_1(T, V, \vec{v} = 0)\left(1 + \frac{1}{2d}\frac{\Gamma\left[\frac{d+2}{z}\right]}{\Gamma\left[\frac{d}{z}\right]}\left(\frac{v\beta}{(\lambda\beta)^{1/z}}\right)^2 + \cdots\right) \,.
\end{aligned}
\tag{134}
$$

This series correctly reproduces the textbook results for $z = 1$ and $z = 2$ respectively, as one can easily check.

The grand-canonical partition function is given by

$$\mathscr{Z}(\mu, T, V, \vec{v}) = \sum_{N=0}^{\infty} e^{\beta\mu N} Z(N, T, V, \vec{v}) = \sum_{N=0}^{\infty} \frac{1}{N!}\left(e^{\beta\mu} Z_1(T, V, \vec{v})\right)^N \,, \tag{135}$$

where $\mu$ is the chemical potential. It follows that

$$\log \mathscr{Z}(\mu, T, V, \vec{v}) = e^{\beta\mu} Z_1(T, V, \vec{v}) \,, \tag{136}$$

and as before, we define the grand potential $\Omega$ as

$$\Omega(\mu, T, V, \vec{v}) = -k_B T \log \mathscr{Z} \,. \tag{137}$$

---

[10]We used the formula $\Gamma(n + \frac{1}{2}) = \frac{(2n)!}{4^n n!}\sqrt{\pi}$.

We first restrict to zero velocities, $\vec{v} = 0$. The grand potential can then be computed explicitly, and the result is

$$\Omega_0 = -\frac{2V}{z}\left(\frac{\sqrt{\pi}}{h}\right)^d \lambda^{-\frac{d}{z}}\frac{\Gamma\left[\frac{d}{z}\right]}{\Gamma\left[\frac{d}{2}\right]}\beta^{-1-\frac{d}{z}}\,e^{\beta\mu}\,. \tag{138}$$

For non-zero velocities, one can again write the grand potential in terms of a power series, using (136) and (134). Taking derivatives then gives all thermodynamic quantities in the grand-canonical ensemble, for instance the kinetic mass density is computed from $P_i = -(\partial\Omega/\partial v^i)_{T,V,\mu} = \rho V v_i$, which at zero velocity is

$$\rho_0 = \frac{2}{zd}\left(\frac{\sqrt{\pi}}{h}\right)^d \frac{\Gamma\left[\frac{d+2}{z}\right]}{\Gamma\left[\frac{d}{2}\right]}\beta(\lambda\beta)^{-\left(\frac{d+2}{z}\right)}e^{\beta\mu}\,. \tag{139}$$

For this result, we explicitly needed the $v$-dependence of the partition function to lowest order in equation (134). Similarly we can compute the pressure from $\Omega = -PV$, and energy from (129). Using the formula for the speed of sound derived in (116) we then find

$$v_s^2 = (d + z)\frac{\Gamma\left[\frac{d}{z}\right]}{\Gamma\left[\frac{d+2}{z}\right]}(k_B T)^{2\left(\frac{z-1}{z}\right)}\lambda^{\frac{2}{z}}\,, \tag{140}$$

which is consistent with (122). Notice that the chemical potential dropped out of this expression. For $z = 1$ and $z = 2$ this formula reproduces the well known results $v_s^2 = \frac{c^2}{d}$ and $v_s^2 = \frac{d+2}{d}\frac{k_B T}{m}$ respectively.

Using these formula, for general $z$, we can write the expansion for the partition function also as an expansion in $v/v_s$:

$$Z_1(T, V, \vec{v}) = Z_1(T, V, \vec{v} = 0)\left(1 + \frac{d+z}{2d}\frac{v^2}{v_s^2} + \frac{1}{8}\frac{(d+z)^2}{d(d+2)}\frac{\Gamma\left[\frac{d}{z}\right]\Gamma\left[\frac{d+4}{z}\right]}{\Gamma\left[\frac{d+2}{z}\right]^2}\frac{v^4}{v_s^4} + \cdots\right). \tag{141}$$

The particle number density is different from the kinetic mass density and given by (at zero velocity)

$$n_0 \equiv \frac{N_0}{V} = e^{\beta\mu}\frac{2}{z}\left(\frac{\sqrt{\pi}}{h}\right)^d\frac{\Gamma\left[\frac{d}{z}\right]}{\Gamma\left[\frac{d}{2}\right]}\left(\frac{k_B T}{\lambda}\right)^{\frac{d}{z}} = e^{\beta\mu}\lambda_{th}^{-d}\,, \tag{142}$$

where we computed $N$ from the relation $N = -(\partial\Omega/\partial\mu)_{(T,V,\vec{v})}$. The relation with the kinetic mass density is

$$\rho_0 = \frac{d+z}{d}\frac{k_B T}{v_s^2}n_0 = \frac{z}{d}\frac{\mathscr{E}_0 + P_0}{v_s^2}\,. \tag{143}$$

The second equality is of course a rewriting of (116) which, using the adiabatic index, can be rewritten as

$$P_0 = \frac{v_s^2}{\gamma}\rho_0\,, \tag{144}$$

which is also familiar from mono-atomic gases. Notice that this not the so-called isothermal sound-speed, in which the temperature is kept fixed and the sound-speed is computed from $(\partial P/\partial\rho)_T$. Instead it is the adiabatic sound speed, in which the entropy is kept fixed while the gas adiabatically expands. The two sound-speeds are known to differ by a multiplicative factor $\gamma^{1/2}$, which is also visible in (144). In other words, adiabatic sound waves move $\gamma^{1/2}$ times faster than isothermal sound waves.

For finite velocity, we can determine the kinetic mass density as a power series, and find to leading order in $v/v_s$

$$\rho = \rho_0 \left( 1 + \frac{1}{2} \frac{(d+z)}{(d+2)} \frac{\Gamma\left[\frac{d}{z}\right]\Gamma\left[\frac{d+4}{z}\right]}{\Gamma\left[\frac{d+2}{z}\right]^2} \frac{v^2}{v_s^2} + \cdots \right) . \tag{145}$$

The number density at finite velocity is given by

$$n = e^{\beta\mu} \frac{Z_1}{V} = n_0 \left( 1 + \frac{d+z}{2d} \frac{v^2}{v_s^2} + \cdots \right) , \tag{146}$$

and for the pressure we find, using (134),

$$P = P_0 \left( 1 + \frac{1}{2d} \frac{\Gamma\left[\frac{d+2}{z}\right]}{\Gamma\left[\frac{d}{z}\right]} \left( \frac{v\beta}{(\lambda\beta)^{1/z}} \right)^2 + \cdots \right) . \tag{147}$$

An equation of state at finite velocity relating $\rho$ and $n$ is harder to read off now. Of course, we still have that (13) holds.

Finally, we mention, as a consistency check, that we have verified the $z$-trace condition on the energy-momentum tensor for any value of the velocity.

**Large $z$-limit**

There is an interesting limit, in which we take the dynamical exponent to be large, keeping

$$\tilde{\lambda} \equiv \lambda^{-1/z} , \tag{148}$$

constant. This is a well defined combination in the large $z$-limit and has dimension of kg m/s. One easily finds that the sound speed grows linearly with $z$ for large $z$, and so does the adiabatic index $\gamma$,

$$v_s^2 \to z \frac{d+2}{d} \left( \frac{k_B T}{\tilde{\lambda}} \right)^2 , \qquad \gamma \to \frac{z}{d} . \tag{149}$$

The kinetic mass density in this limit is related to the number density, at zero velocity, by

$$\rho_0 = \frac{\beta \tilde{\lambda}^2}{d+2} n_0 , \tag{150}$$

and one can check that the ideal gas law $P_0 = (k_B T)n_0$ is still satisfied.

For finite velocity, it is still difficult to sum the series even in the large $z$-limit. But we can work in perturbation theory, and find the velocity dependent corrections to e.g. the pressure,

$$P = P_0 \left[ 1 + \frac{\beta^2 \tilde{\lambda}^2 v^2}{2(d+2)} + \cdots \right] , \tag{151}$$

consistent with taking the limit on (147).

## 4.2 Quantum gases

The previously discussed ideal gas is at low density. For higher densities, spin statistics influences the allowed occupation of states. For a given state, labelled by momentum $\vec{k}$, the grand canonical partition function is given by

$$\mathcal{Z}_{\vec{k}} = \sum_{n_{\vec{k}}} e^{-n_{\vec{k}}\beta\left(E_1(\vec{k})-\vec{v}\cdot\vec{k}-\mu\right)} . \tag{152}$$

Here $n_{\vec{k}}$ denotes the occupation number, and $E_1$ is the energy for a single particle. For bosons this number spans $n_{\vec{k}} = \{0, 1, 2, ..\}$, whereas for fermions this is restricted to $n_{\vec{k}} = \{0, 1\}$. We can perform the sum and find

$$\mathscr{Z}_{B/F,\vec{k}} = \left(1 \mp e^{-\beta(H_1(\vec{k}) - \vec{v}\cdot\vec{k} - \mu)}\right)^{\mp 1}, \tag{153}$$

where bosons (fermions) correspond to the upper (lower) sign. To avoid singularities, we restrict to $\mu < 0$, for bosons, in other words $0 < e^{\beta\mu} < 1$. For fermions we allow $0 < e^{\beta\mu} < \infty$. For non-interacting particles, multi-particle states all contribute to the grand canonical partition function in a factorized way, and we get

$$\mathscr{Z}_{B/F} = \prod_{\vec{k}} \mathscr{Z}_{B/F,\vec{k}}^{2s+1}, \tag{154}$$

where $s$ denotes the spin multiplicity. The grand potential is then obtained by

$$\begin{aligned}
\Omega_{B/F} &\equiv -\frac{1}{\beta} \log\left(\mathscr{Z}_{B/F}\right) \\
&= \pm(2s+1)\frac{1}{\beta} \sum_{\vec{k}} \log\left(1 \mp e^{-(\beta H_1(\vec{k}) - \beta\vec{v}\cdot\vec{k} - \beta\mu)}\right) \\
&\approx \pm(2s+1)\frac{1}{\beta}\frac{V}{h^d} \int d^d\vec{k}\, \log\left(1 \mp e^{-(\beta H_1(\vec{k}) - \beta\vec{v}\cdot\vec{k} - \beta\mu)}\right),
\end{aligned} \tag{155}$$

where in the third line we approximated the sum by an integral, which is valid as long as $\lambda_{th} \ll L \equiv V^{1/d}$. This requirement will fail for low temperatures, since $\lambda_{th}^d \sim T^{-d/z}$. We will discuss this later.

Evaluating the integral in (155) we obtain

$$\begin{aligned}
\Omega_{B/F} &= \mp(2s+1)V\lambda_{th}^{-d}\frac{\Gamma\left[\frac{d}{2}\right]}{\Gamma\left[\frac{d}{z}\right]}\frac{1}{\beta}\sum_{n=0}^{\infty}\frac{(\beta v)^{2n}}{(2n)!}\frac{(2n-1)!!}{2^n}\frac{\Gamma\left[\frac{d+2n}{z}\right]}{\Gamma\left[\frac{d}{2}+n\right]}(\lambda\beta)^{-\frac{2n}{z}}\mathrm{Li}_{\frac{d+2n}{z}+1-2n}\left[\pm e^{\beta\mu}\right] \\
&= \Omega_{B/F}(T, V, \mu, \vec{v} = 0)\left(1 + \frac{1}{2d}\frac{\Gamma\left[\frac{d+2}{z}\right]}{\Gamma\left[\frac{d}{z}\right]}\left(\frac{\beta v}{(\lambda\beta)^{\frac{1}{z}}}\right)^2 \frac{\mathrm{Li}_{\frac{d+2}{z}-1}\left[\pm e^{\beta\mu}\right]}{\mathrm{Li}_{\frac{d}{z}+1}\left[\pm e^{\beta\mu}\right]} + \cdots\right),
\end{aligned} \tag{156}$$

where $\mathrm{Li}_n(z)$ denotes a polylogarithm and where

$$\Omega_{B/F}(T, V, \mu, \vec{v} = 0) = \mp(2s+1)V\lambda_{th}^{-d}\frac{1}{\beta}\mathrm{Li}_{\frac{d}{z}+1}\left[\pm e^{\beta\mu}\right]. \tag{157}$$

For $e^{\beta\mu} \ll 1$, so low density or high temperature, the result in (156) coincides,[11] as can be seen from (155), with the result from the Boltzmann gas from (134), up to an overall factor of $(2s+1)$. The particle number and pressure (from which the internal energy density can be obtained by using the $z$-deformed trace condition) can be computed using thermodynamic identities related to the grand potential, for which we for simplicity consider $\vec{v} = 0$,

$$N_{B/F,0} = \pm(2s+1)V\lambda_{th}^{-d}\mathrm{Li}_{\frac{d}{z}}\left[\pm e^{\beta\mu}\right], \tag{158}$$

$$P_{B/F,0}V = N_{B/F,0}k_B T\frac{\mathrm{Li}_{\frac{d}{z}+1}\left[\pm e^{\beta\mu}\right]}{\mathrm{Li}_{\frac{d}{z}}\left[\pm e^{\beta\mu}\right]}. \tag{159}$$

---

[11]Using that $\mathrm{Li}_n(x) \approx x$ for $x \ll 1$.

Notice that the last line can be regarded as a deformation of the ideal gas law.

We shall now compute $\hat{C}_V^{B/F}$ and $\hat{C}_P^{B/F}$. We use (158) to compute $\partial_T \mu$ at fixed $N$ and $V$. This step is needed in order to find a closed expression. We motivate the finite $N$ by having switched to a canonical setup. In the thermodynamic limit of large $N$ the canonical and grand canonical ensembles however coincide. The $\hat{C}_V^{B/F}$ becomes, defined by (128),

$$\hat{C}_V^{B/F} = \frac{d}{z}\left[\left(1 + \frac{d}{z}\right)\frac{\mathrm{Li}_{\frac{d}{z}+1}(\pm e^{\beta\mu})}{\mathrm{Li}_{\frac{d}{z}}(\pm e^{\beta\mu})} - \frac{d}{z}\frac{\mathrm{Li}_{\frac{d}{z}}(\pm e^{\beta\mu})}{\mathrm{Li}_{\frac{d}{z}-1}(\pm e^{\beta\mu})}\right], \tag{160}$$

In combination with equations (158) and (159) we compute, using definition (130),

$$\hat{C}_P^{B/F} = \left(1 + \frac{d}{z}\right)\frac{\mathrm{Li}_{\frac{d}{z}+1}(\pm e^{\beta\mu})\mathrm{Li}_{\frac{d}{z}-1}(\pm e^{\beta\mu})}{\left[\mathrm{Li}_{\frac{d}{z}}(\pm e^{\beta\mu})\right]^2}\left[\left(1 + \frac{d}{z}\right)\frac{\mathrm{Li}_{\frac{d}{z}+1}(\pm e^{\beta\mu})}{\mathrm{Li}_{\frac{d}{z}}(\pm e^{\beta\mu})} - \frac{d}{z}\frac{\mathrm{Li}_{\frac{d}{z}}(\pm e^{\beta\mu})}{\mathrm{Li}_{\frac{d}{z}-1}(\pm e^{\beta\mu})}\right]. \tag{161}$$

The results for $d = 3$ and $z = 2$ for $\hat{C}_P$ and $\hat{C}_V$ agree with the findings of [33] for the bosonic case. For the fermionic case we check the low temperature behavior of $\hat{C}_V^F$, using the Sommerfeld expansion

$$-\mathrm{Li}_n(-e^{\beta\mu}) = \frac{(\beta\mu)^n}{\Gamma(n+1)}\left(1 + \frac{\pi^2}{6}\frac{n(n-1)}{(\beta\mu)^2} + \dots\right), \tag{162}$$

which results into

$$\hat{C}_V^F = \frac{d}{z}\frac{\pi^2}{3}\frac{T}{T_F}. \tag{163}$$

Here $T_F$ is the Fermi temperature, defined via the Fermi energy $E_F$ that can be found using (158) and taking $T \to 0$ and $\mu(T \to 0) = E_F$,

$$T_F \equiv E_F/k_B = \frac{\lambda\hbar^z}{k_B}\left[\frac{nd2^{d-1}\pi^{d/2}}{2s+1}\Gamma[d/2]\right]^{\frac{z}{d}}. \tag{164}$$

The expressions above coincide with the known results for $d = 3$, $z = 2$ and $s = 1/2$.

For quantum gases, the adiabatic index is not defined by the ratio of the heat capacities. Instead one defines it via the relation $PV^\gamma = constant$, and, as for the classical case, it is fixed by Lifshitz scale symmetry to be

$$\gamma = 1 + \frac{z}{d}. \tag{165}$$

It is well known for the Bose gas that there exists the possibility to form a Bose-Einstein condensate, in which the ground state becomes a highly occupied state. We can see this by considering (158). As we keep $N$ and $V$ constant and lower $T$ towards zero, the value of the polylog should increase in order to compensate for the decreasing value of $\lambda_{th}^{-d} \sim T^{d/z}$. For $d > z$, however, the polylog is bounded from above,

$$\mathrm{Li}_n(1) = \zeta(n), \tag{166}$$

for $n > 1$. For $n \leq 1$ the function diverges. Thus, as we arrive at some $T = T_c > 0$ the polylog attains its maximal value. The value of $T_c$, which is computed using (158) and $e^{\beta\mu} \to 1$, is

$$k_B T_c = \lambda\left[\frac{N}{V}\frac{z}{2}\left(\frac{h}{\sqrt{\pi}}\right)^d \frac{1}{\zeta\left(\frac{d}{z}\right)\Gamma\left[\frac{d}{z}\right]}\right]^{z/d}. \tag{167}$$

Using the expression for the particle number density, we can rewrite this equation as

$$\frac{T}{T_c} = \left[\frac{\zeta(\frac{d}{z})}{\text{Li}_{\frac{d}{z}}(e^{\beta\mu})}\right]^{z/d}, \tag{168}$$

which can be used to trade $\beta\mu$ for $T/T_c$.

By approximating the sum of discrete momenta with an integral, we fail to take into account contributions of the ground state properly for $T < T_c$. From [32], it is known that no Bose-Einstein condensation takes place for $z \geq d$. For $z \to d$, we get from $\zeta(1) = \infty$ that $T_c \to 0$. The case $z = d$ is also special from other points of view, e.g. in the relaxation behavior of Lifshitz theories at strong coupling, see e.g. [34].

We can remedy failing to take into account the ground state, by explicitly taking

$$\begin{aligned}
\Omega^{T \leq T_c} &= -\frac{1}{\beta}\log(\mathscr{Z}_B) \\
&= (2s+1)\frac{1}{\beta}\sum_{\vec{k}\neq 0}\log\left(1 - e^{-(\beta H_1(\vec{k})+\beta\vec{v}\cdot\vec{k}-\beta\mu)}\right) + (2s+1)\frac{1}{\beta}\log\left(1 - e^{-(\beta H_1(\vec{k})-\beta\vec{v}\cdot\vec{k}-\beta\mu)}\right)\Big|_{\vec{k}=0} \\
&\approx (2s+1)\frac{1}{\beta}\frac{V}{h^d}\int d^d\vec{k}\,\log\left(1 - e^{-(\beta H_1(\vec{k})-\beta\vec{v}\cdot\vec{k}-\beta\mu)}\right) + (2s+1)\frac{1}{\beta}\log\left(1 - e^{\beta\mu}\right).
\end{aligned} \tag{169}$$

If we now compute $N$ for $T \leq T_c$, this results in

$$N = (2s+1)V\lambda_{th}^{-d}\text{Li}_{\frac{d}{z}}[e^{\beta\mu}] + (2s+1)\frac{e^{\beta\mu}}{1 - e^{\beta\mu}}. \tag{170}$$

For large $N$ as $T \leq T_c$ we find that $e^{\beta\mu} \approx 1 - (2s+1)/N$ in order for equation (170) to be self-consistent at leading order. If we define $N_0 \equiv (2s+1)e^{\beta\mu}/(1 - e^{\beta\mu})$ as the particles in the groundstate we can derive using the formula above

$$\begin{aligned}
\frac{N_0}{N} &= 1 - (2s+1)\frac{V}{N}\lambda_{th}^{-d}\zeta\left(\frac{d}{z}\right) + \mathcal{O}(1/N^2) \\
&= 1 - \left(\frac{T}{T_c}\right)^{d/z},
\end{aligned} \tag{171}$$

where in the first line we used $T_c$ as defined in (167).

Using that $e^{\beta\mu} \approx 1 - (2s+1)/N$ for $T < T_c$ we can simplify $\Omega^{T \leq T_c}$ to

$$\Omega^{T \leq T_c} = -(2s+1)V\lambda_{th}^{-d}\frac{1}{\beta}\zeta\left(\frac{d}{z}+1\right) + (2s+1)\frac{1}{\beta}\log\left(\frac{2s+1}{N}\right) + \mathcal{O}(1/N). \tag{172}$$

We can now compute the pressure from

$$P^{T \leq T_c} = \frac{N}{V}k_B T\frac{\zeta\left(\frac{d}{z}+1\right)}{\zeta\left(\frac{d}{z}\right)}\left(\frac{T}{T_c}\right)^{d/z}, \tag{173}$$

where in the second line we used (171). Applying the $z$-trace identity (at $\vec{v} = 0$) we obtain the internal energy

$$\mathscr{E}^{T \leq T_c} = \frac{N}{V}k_B T\frac{\zeta\left(\frac{d}{z}+1\right)}{\zeta\left(\frac{d}{z}\right)}\left(\frac{T}{T_c}\right)^{d/z}. \tag{174}$$

The heat capacity is given by

$$\hat{C}_V^{T \leq T_c} = \frac{d}{z}\left(1 + \frac{d}{z}\right)\frac{\zeta\left(\frac{d}{z}+1\right)}{\zeta\left(\frac{d}{z}\right)}\left(\frac{T}{T_c}\right)^{d/z}. \tag{175}$$

These results agree with [35], in which it is also noted that when comparing (175) and (160) near $T = T_c$ is only continuous when $z < d \leq 2z$. The first inequality follows from the requirement of having a $T_c$ in the first place, the second follows from (166). When $d > 2z$ then $C_V$ is discontinuous across $T_c$. This is to be expected from the kinks in the internal energy density $\tilde{\mathcal{E}}$.

The isobaric heat capacity $\hat{C}_P^{T \leq T_c}$ becomes undefined since fixing pressure and deriving with respect to temperature is inconsistent, as can be seen from writing out $P^{T \leq T_c}$. However, $\hat{C}_P$ above $T_c$, as approaching $T_c$, diverges if $d \leq 2z$. Beyond that point, for $d > 2z$, $\hat{C}_P$ obtains some finite value.

In order to compute the speed of sound, we need $\rho_{B/F}$. The result is

$$\rho_{B/F,0} = -\frac{\Omega_{B/F}(T, V, \mu, \vec{v} = 0)}{V} \frac{1}{d} \frac{\Gamma\left[\frac{d+2}{z}\right]}{\Gamma\left[\frac{d}{z}\right]} \left(\frac{\beta}{(\lambda\beta)^{\frac{1}{z}}}\right)^2 \frac{\text{Li}_{\frac{d+2}{z}-1}\left[\pm e^{\beta\mu}\right]}{\text{Li}_{\frac{d}{z}+1}\left[\pm e^{\beta\mu}\right]}, \tag{176}$$

using the same formula as described above (116), supplied with (156). The superscript zero denotes that $v = 0$. Using the formula for the speed of sound (116), we end up with

$$v_{B/F}^2 = v_{\text{Boltzmann}}^2 \frac{\text{Li}_{\frac{d}{z}+1}\left[\pm e^{\beta\mu}\right]}{\text{Li}_{\frac{d+2}{z}-1}\left[\pm e^{\beta\mu}\right]}. \tag{177}$$

Contrary to the speed of sound of the Boltzmann gas (140), the dependence on the chemical potential does not drop out of the expression. It is clear that for $z = 1$, we find $v_{B/F}^2 = v_{\text{Boltzmann}}^2$. When $e^{\beta\mu} \ll 1$, we find $v_{B/F}^2 = v_{\text{Boltzmann}}^2$. Comparing the results for the speed of sound in the Boltzmann gas to the quantum gas we furthermore notice that

$$0 \leq v_B^2 \leq v_{\text{Boltzmann}}^2 \leq v_F^2. \tag{178}$$

In contrast to $C_V$ and $C_P$, the definition of $\rho$ does not involve derivatives of the grand potential with respect to temperature. One can therefore take the $T > T_c$ expression and simply replace, $e^{\beta\mu} \approx 1 - (2s + 1)/N$ with large $N$, without the risk of glossing over terms. This leads to the following expression for the speed of sound for the gas of particles (the contributions of the particles in the condensate are suppressed by large $N$)

$$\left(v_B^{T \leq T_c}\right)^2 = v_{\text{Boltzmann}}^2 \frac{\zeta\left[\frac{d}{z} + 1\right]}{\zeta\left[\frac{d+2}{z} - 1\right]}, \tag{179}$$

with the requirement that $d > 2(z - 1)$. For $z < d \leq 2(z - 1)$ we find that the speed of sound goes to zero below $T_c$. For $d = 3$ and $z = 2$ we reproduce the known result given in for instance [35]. We illustrate the behavior of the speed of sound in Figures 1 and 2.

When restricting to $d \leq 3$ above $T_c$, we find new results for $z > 2$. In the case of $T \leq T_c$ we are restricted to $d > z$, so when considering in addition to $d \leq 3$ we are forced to $z < 3$. This means that for integer values of $z$ we do not find results for $z > 2$ under $T_c$. In other words, for $d \leq 3$ we only have novel (i.e. $z > 2$) predictions above $T_c$. Notice that for $d = 3$, the allowed range of $z$ is $5/2 \leq z < 3$. We stress that all these results are for ideal gases. It would be interesting to see what the effects of adding interactions would be.

## 5 Discussion and outlook

With the general theory of perfect fluids for non-boost invariant systems presented in this paper, a natural and important next step is to consider the derivative expansion of the conserved

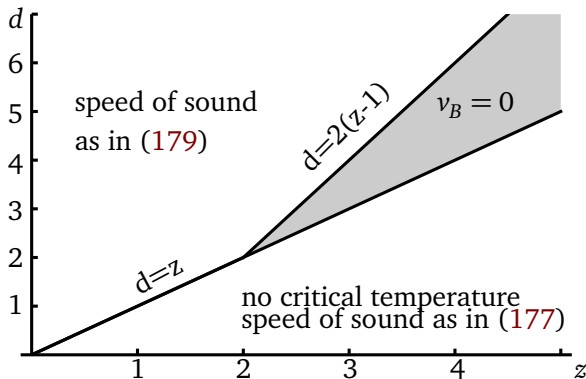

Figure 1: We give a representation of the speed of sound of the Bose gas for $T \leq T_c$. There are three distinct regions: speed of sound as in (179), speed of sound is zero $v_B = 0$, and no critical temperature and thus speed of sound as in (177).

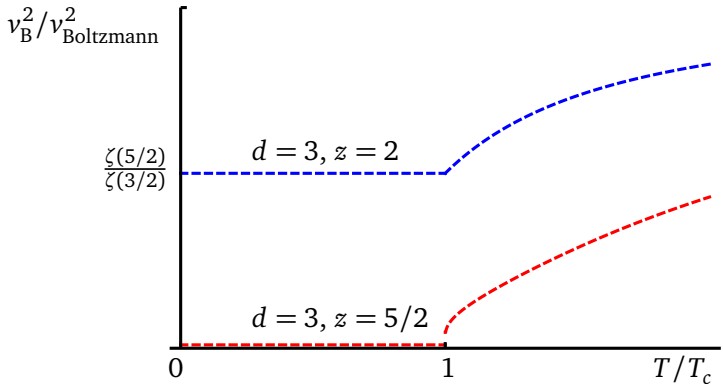

Figure 2: We present two different situations for the speed of sound of the Bose gas. The upper line ($d = 3$ and $z = 2$) corresponds to a finite speed of sound in the condensate phase (for $T > 0$), whereas the lower line ($d = 3$ and $z = 5/2$) corresponds to a speed of sound that is equal to zero anywhere in the condensate phase.

currents in terms of relevant fluid variables and study transport corrections. The case of linear perturbations around a global thermodynamic system at rest and corresponding hydrodynamic modes will appear in [3], while a more general first order analysis at non-zero finite velocity will be treated in [4]. Beyond this, there are a host of potentially interesting aspects to consider further, such as momentum dissipation, turbulence, shock waves, surface phenomena and many other analogues of well-studied aspects of boost-invariant hydrodynamics.

We have discussed in detail the case of a free gas of Lifshitz particles as an illustration of general perfect fluids. Here, an interesting follow-up is to consider interacting Lifshitz particles by adding potentials that depend on the intra-particle distances. An example of such a system is given by the $N$-particle Hamiltonian

$$H = \lambda \sum_{i=1}^{N} (\vec{p}_i^{\,2})^{\frac{z}{2}} + \sum_{i \neq j}^{N} \frac{g}{|\vec{x}_i - \vec{x}_j|^z} \,, \tag{180}$$

with coupling constant $g$.

Another direction pertains the study of field theory models for cases with non-integer $z$. In this connection, the results of [36], which considers ripplons on domain walls between two different superfluids, may be relevant. By integrating out the gapless modes of the superfluids one finds non-local kinetic terms, which in Fourier space correspond to dispersion relations

with fractional powers. This suggests more generally that non-integer $z$ might imply non-local kinetic terms.

It will also be worthwhile to further develop a holographic fluid/gravity correspondence for field theory systems with Lifshitz symmetry[12] (i.e. without boost symmetry) building on our results. In this connection, we note that for theories with $z = 2$ Schrödinger symmetries a version of (conformal) non-relativistic fluid/gravity correspondence was developed in Refs. [38–41]. Specific examples of Lifshitz hydrodynamics and their holographic description have subsequently been considered in Refs. [42–44] and aspects of non-Galilean invariant hydrodynamics have also been treated in the context of momentum dissipation in holography (see e.g. [2, 45, 46]). Hydrodynamics of certain Lifshitz theories with $z = 2$ has furthermore been holographically studied in a bulk Einstein-Maxwell-dilaton (EMD) model [19, 47] in which there is an extra bulk $U(1)$ symmetry. Since the dilaton runs logarithmically close to the boundary, there is a new scaling exponent on top of the dynamical exponent $z$ in these models. For the Einstein-Proca-dilaton (EPD) model, 4-dimensional $z = 2$ Lifshitz black branes with nonzero linear momentum have been constructed [5], providing a holographic realization of a non-boost invariant fluid without a U(1) symmetry. This is another interesting setting in which one can holographically compute equations of state and first-order transport coefficients for the systems considered in this paper. Finally, hydrodynamics for theories with Hořava-Lifshitz gravity in the bulk [48, 49] may also be a relevant case to consider.

To conclude, it will be very interesting to examine experimental consequences of the results of this paper, and make an inventory of the type of fluid systems in nature that are homogeneous and isotropic, but for which boost symmetry is broken. In particular, we emphasize that the speed of sound in such situations is given by the general expression (112) which presents a potentially measurable quantity. We also stress that our theory leads to specific corrections to the known form of the Euler equation for perfect fluids (see the generalized Euler equation (33)) which may be observable in hydrodynamic fluid experiments. In particular, the correction terms involves the kinetic mass density $\rho(T, \mu, v^2)$, for which one could imagine determining the velocity profile (along with the $T, \mu$ dependence) using experimental data. Moreover, we note that as fluids play an essential role in cosmology, there could be useful applications in that arena as well.

## Acknowledgements

We thank Alexander Abanov, Rembert Duine, Blaise Goutéraux, Sašo Grozdanov, Kristan Jensen, Elias Kiritsis, Koenraad Schalm and Henk Stoof for useful discussions. The work of NO is supported in part by the project "Towards a deeper understanding of black holes with non-relativistic holography" of the Independent Research Fund Denmark (grant number DFF-6108-00340). All the authors thank Nordita for hospitality and support during the 2016 workshop "Black Holes and Emergent Spacetime". JH and NO gratefully acknowledge support from the Simons Center for Geometry and Physics, Stony Brook University at which some of the research for this paper was performed during the 2017 workshop "Applied Newton-Cartan Geometry". JH acknowledges hospitality of Niels Bohr Institute and NO acknowledges hospitality of University of Amsterdam and University of Utrecht during part of this work. This work was further supported by the Netherlands Organisation for Scientific Research (NWO) under the VICI grant 680-47-603, and the Delta-Institute for Theoretical Physics (D-ITP) that is funded by the Dutch Ministry of Education, Culture and Science (OCW).

---

[12]See Ref. [37] for a review on Lifshitz holography.

# A Speed of sound for a boosted fluid

In this appendix, we study the speed of sound propagating in a fluid which is moving at some velocity $v_0^i \neq 0$, contrasting the case we have studied up to now where we considered a fluid at rest, i.e. $v_0^i = 0$. This computation, although more lengthy, is carried out analogously to the prior computation where $v_0^i = 0$. The linearized fluid equations can be written in the following form

$$\rho_0(\partial_t + v_0^i\partial_i)\delta v^j + v_0^j(\partial_t + v_0^i\partial_i)\delta\rho + \rho_0 v_0^j\partial_i\delta v^i + \partial_j\delta P = 0\,, \tag{181}$$

$$(\partial_t + v_0^i\partial_i)\delta\tilde{\mathscr{E}} + (\tilde{\mathscr{E}}_0 + P_0)\partial_i\delta v^i + \rho_0 v_0^j(\partial_t + v_0^i\partial_i)\delta v^j = 0\,, \tag{182}$$

$$(\partial_t + v_0^i\partial_i)\delta n + n_0\partial_i\delta v^i = 0\,. \tag{183}$$

The last equation can be ignored if we do not consider an additional conserved current. In order to find the speed of sound in the moving fluid we express $P$ and $\rho$ in terms of $\tilde{\mathscr{E}}$, $v^2$, and $n$.

Let us perform a Galilean boost to a coordinate system $(t', x'^i)$ where $\partial_{t'} = \partial_t + v_0^i\partial_i$ and $\partial_i' = \partial_i$. Next we Fourier transform $\delta\tilde{\mathscr{E}} = e^{-i\omega't'+ik'^ix'^i}\delta\tilde{\mathscr{E}}(\omega', k')$. Let us define $\delta v^2 = 2v_0^i\delta v^i$ and $k'^i\delta v^i = k'\delta v_\parallel$. There are $d-2$ modes with zero frequency $\omega' = 0$ that are orthogonal to both $v_0^i$ and $k'^i$. By contracting the Fourier transformed version of (181) once with $v_0^i$ and once with $k'^i$ we obtain together with the two other equations, the following four coupled equations

$$
\begin{aligned}
0 ={}& -\frac{1}{2}\rho_0\omega'\delta v^2 - v_0^2\omega'\left[\left(\frac{\partial\rho_0}{\partial\tilde{\mathscr{E}}_0}\right)_{n_0,v_0^2}\delta\tilde{\mathscr{E}} + \left(\frac{\partial\rho_0}{\partial n_0}\right)_{\tilde{\mathscr{E}}_0,v_0^2}\delta n + \left(\frac{\partial\rho_0}{\partial v_0^2}\right)_{\tilde{\mathscr{E}}_0,n_0}\delta v^2\right] \\
&+ \rho_0 v_0^2 k'\delta v_\parallel + v_0^i k'^i\left[\left(\frac{\partial P_0}{\partial\tilde{\mathscr{E}}_0}\right)_{n_0,v_0^2}\delta\tilde{\mathscr{E}} + \left(\frac{\partial P_0}{\partial n_0}\right)_{\tilde{\mathscr{E}}_0,v_0^2}\delta n + \left(\frac{\partial P_0}{\partial v_0^2}\right)_{\tilde{\mathscr{E}}_0,n_0}\delta v^2\right]\,, \quad (184)
\end{aligned}
$$

$$
\begin{aligned}
0 ={}& -\rho_0\omega' k'\delta v_\parallel - v_0^i k'^i\omega'\left[\left(\frac{\partial\rho_0}{\partial\tilde{\mathscr{E}}_0}\right)_{n_0,v_0^2}\delta\tilde{\mathscr{E}} + \left(\frac{\partial\rho_0}{\partial n_0}\right)_{\tilde{\mathscr{E}}_0,v_0^2}\delta n + \left(\frac{\partial\rho_0}{\partial v_0^2}\right)_{\tilde{\mathscr{E}}_0,n_0}\delta v^2\right] \\
&+ \rho_0 v_0^i k'^i k'\delta v_\parallel + k'^2\left[\left(\frac{\partial P_0}{\partial\tilde{\mathscr{E}}_0}\right)_{n_0,v_0^2}\delta\tilde{\mathscr{E}} + \left(\frac{\partial P_0}{\partial n_0}\right)_{\tilde{\mathscr{E}}_0,v_0^2}\delta n + \left(\frac{\partial P_0}{\partial v_0^2}\right)_{\tilde{\mathscr{E}}_0,n_0}\delta v^2\right]\,, \quad (185)
\end{aligned}
$$

$$0 = -\omega'\delta\tilde{\mathscr{E}} + (\tilde{\mathscr{E}}_0 + P_0)k'\delta v_\parallel - \frac{1}{2}\rho_0\omega'\delta v^2\,, \tag{186}$$

$$0 = -\omega'\delta n + n_0 k'\delta v_\parallel\,. \tag{187}$$

By using the last two equations to eliminate $\delta v^2$ and $\delta v_\parallel$ in favor of $\delta\tilde{\mathscr{E}}$ and $\delta n$ we obtain two coupled equations that take the following form

$$\begin{pmatrix} A & B \\ C & D \end{pmatrix}\begin{pmatrix} \delta\tilde{\mathscr{E}} \\ \delta n \end{pmatrix} = 0\,, \tag{188}$$

where

$$
\begin{aligned}
A = {}& \omega' + \frac{2}{\rho_0} v_0^2 \omega' \left( \frac{\partial \rho_0}{\partial v_0^2} \right)_{\tilde{\mathcal{E}}_0, n_0} - \frac{2}{\rho_0} v_0^i k'^i \left( \frac{\partial P_0}{\partial v_0^2} \right)_{\tilde{\mathcal{E}}_0, n_0} - v_0^2 \omega' \left( \frac{\partial \rho_0}{\partial \tilde{\mathcal{E}}_0} \right)_{n_0, v_0^2} \\
& + v_0^i k'^i \left( \frac{\partial P_0}{\partial \tilde{\mathcal{E}}_0} \right)_{n_0, v_0^2},
\end{aligned}
\tag{189}
$$

$$
\begin{aligned}
B = {}& -\frac{\tilde{\mathcal{E}}_0 + P_0}{n_0} \omega' - \frac{2}{\rho_0} v_0^2 \frac{\tilde{\mathcal{E}}_0 + P_0}{n_0} \omega' \left( \frac{\partial \rho_0}{\partial v_0^2} \right)_{\tilde{\mathcal{E}}_0, n_0} + \frac{2}{\rho_0} \frac{\tilde{\mathcal{E}}_0 + P_0}{n_0} v_0^i k'^i \left( \frac{\partial P_0}{\partial v_0^2} \right)_{\tilde{\mathcal{E}}_0, n_0} \\
& - v_0^2 \omega' \left( \frac{\partial \rho_0}{\partial n_0} \right)_{\tilde{\mathcal{E}}_0, v_0^2} + v_0^i k'^i \left( \frac{\partial P_0}{\partial n_0} \right)_{\tilde{\mathcal{E}}_0, v_0^2} + \frac{\rho_0}{n_0} v_0^2 \omega',
\end{aligned}
\tag{190}
$$

$$
\begin{aligned}
C = {}& \frac{2}{\rho_0} v_0^i k'^i \omega' \left( \frac{\partial \rho_0}{\partial v_0^2} \right)_{\tilde{\mathcal{E}}_0, n_0} - \frac{2}{\rho_0} k'^2 \left( \frac{\partial P_0}{\partial v_0^2} \right)_{\tilde{\mathcal{E}}_0, n_0} - v_0^i k'^i \omega' \left( \frac{\partial \rho_0}{\partial \tilde{\mathcal{E}}_0} \right)_{n_0, v_0^2} \\
& + k'^2 \left( \frac{\partial P_0}{\partial \tilde{\mathcal{E}}_0} \right)_{n_0, v_0^2},
\end{aligned}
\tag{191}
$$

$$
\begin{aligned}
D = {}& -\frac{2}{\rho_0} \frac{\tilde{\mathcal{E}}_0 + P_0}{n_0} v_0^i k'^i \omega' \left( \frac{\partial \rho_0}{\partial v_0^2} \right)_{\tilde{\mathcal{E}}_0, n_0} + \frac{2}{\rho_0} \frac{\tilde{\mathcal{E}}_0 + P_0}{n_0} k'^2 \left( \frac{\partial P_0}{\partial v_0^2} \right)_{\tilde{\mathcal{E}}_0, n_0} \\
& - v_0^i k'^i \omega' \left( \frac{\partial \rho_0}{\partial n_0} \right)_{\tilde{\mathcal{E}}_0, v_0^2} + k'^2 \left( \frac{\partial P_0}{\partial n_0} \right)_{\tilde{\mathcal{E}}_0, v_0^2} - \frac{\rho_0}{n_0} \omega'^2 + \frac{\rho_0}{n_0} v_0^i k'^i \omega'.
\end{aligned}
\tag{192}
$$

Let us define

$$
V = \frac{\tilde{\mathcal{E}}_0 + P_0}{\rho_0} \left( \frac{\partial P_0}{\partial \tilde{\mathcal{E}}_0} \right)_{n_0, v_0^2} + \frac{n_0}{\rho_0} \left( \frac{\partial P_0}{\partial n_0} \right)_{\tilde{\mathcal{E}}_0, v_0^2} = \frac{n_0}{\rho_0} \left( \frac{\partial P_0}{\partial n_0} \right)_{\frac{s_0}{n_0}, v_0^2},
\tag{193}
$$

$$
X = 1 + \frac{\tilde{\mathcal{E}}_0 + P_0}{\rho_0} \left( \frac{\partial \rho_0}{\partial \tilde{\mathcal{E}}_0} \right)_{n_0, v_0^2} + \frac{n_0}{\rho_0} \left( \frac{\partial \rho_0}{\partial n_0} \right)_{\tilde{\mathcal{E}}_0, v_0^2} = 1 + \frac{n_0}{\rho_0} \left( \frac{\partial \rho_0}{\partial n_0} \right)_{\frac{s_0}{n_0}, v_0^2},
\tag{194}
$$

$$
Y = \left( \frac{\partial P_0}{\partial \tilde{\mathcal{E}}_0} \right)_{n_0, v_0^2} - \frac{2}{\rho_0} \left( \frac{\partial P_0}{\partial v_0^2} \right)_{\tilde{\mathcal{E}}_0, n_0} = -\frac{2}{\rho_0} \left( \frac{\partial P_0}{\partial v_0^2} \right)_{\frac{s_0}{n_0}, n_0},
\tag{195}
$$

$$
Z = \left( \frac{\partial \rho_0}{\partial \tilde{\mathcal{E}}_0} \right)_{n_0, v_0^2} - \frac{2}{\rho_0} \left( \frac{\partial \rho_0}{\partial v_0^2} \right)_{\tilde{\mathcal{E}}_0, n_0} = -\frac{2}{\rho_0} \left( \frac{\partial \rho_0}{\partial v_0^2} \right)_{\frac{s_0}{n_0}, n_0}.
\tag{196}
$$

The second equalities can be obtained by replacing the variable $\tilde{\mathcal{E}}_0$ by $s_0/n_0$. The equations (188) have a nontrivial solution if and only if $AD - BC = 0$. One solution is for $\omega' = 0$ and the other solution obeys

$$
\left( 1 - v_0^2 Z \right) \omega'^2 + (X + Y) v_0^i k'^i \omega' + (VZ - XY) \left( v_0^2 k'^2 - (v_0^i k'^i)^2 \right) - V k'^2 = 0.
\tag{197}
$$

This quadratic equation for $\omega'$ has two solutions that are related by sending $v_0^i \to -v_0^i$ and $\omega' \to -\omega'$. The equation $AD - BC = 0$ thus has one zero frequency mode and one sound mode. Since $\delta v^2$ and $\delta v_\parallel$ are related to $\delta \tilde{\mathcal{E}}$ and $\delta n$ via a linear transformation (see equations (186) and (187)) it follows that the linear equations satisfied by $\delta v^2$ and $\delta v_\parallel$ again only have a nontrivial solution if and only if $AD - BC = 0$ so that the solutions to this equation have multiplicity two. Before performing the Fourier transformation we performed a Galilean boost so that $\partial_{t'} = \partial_t + v_0^i \partial_i$ and $\partial_i' = \partial_i$. This means that in LAB frame we have $\omega = \omega' + v_0^i k'^i$ and $k^i = k'^i$. The LAB frame speed of sound $v_s$ is then given by $\omega = v_s k$. If we define the angle $\theta$

via $v_0^i k^i = v_0 k \cos \theta$ then the speed of sound is

$$
\begin{aligned}
v_s & = -\left( \frac{X+Y}{2(1-v_0^2 Z)} - 1 \right) v_0 \cos \theta \\
& + \frac{1}{2(1-v_0^2 Z)} \left[ (X+Y)^2 v_0^2 \cos^2 \theta + 4(1-v_0^2 Z)(VZ - XY) v_0^2 \cos^2 \theta \right. \\
& \left. + 4V(1-v_0^2 Z)^2 + 4v_0^2 (1-v_0^2 Z) XY \right]^{1/2} ,
\end{aligned}
\tag{198}
$$

where we chose the sign so that $v_s = V^{1/2}$ for $v_0 = 0$.

We will now show that this formula for the speed of sound gives the expected expressions for the Bargmann and Lorentz invariant cases. Starting with the Bargmann case we have $\rho_0 = n_0$ and $P_0 = P_0(\hat{\mathscr{E}}_0, n_0)$ where $\hat{\mathscr{E}}_0 = \tilde{\mathscr{E}}_0 + \frac{1}{2} n_0 v_0^2$. It follows that $X = Y = Z = 0$, so that

$$
v_s = v_0 \cos \theta + \tilde{v}_s ,
\tag{199}
$$

where

$$
\tilde{v}_s^2 = V = \frac{\hat{\mathscr{E}}_0 + P_0}{n_0} \left( \frac{\partial P_0}{\partial \hat{\mathscr{E}}_0} \right)_{n_0} + \left( \frac{\partial P_0}{\partial n_0} \right)_{\hat{\mathscr{E}}_0} = \left( \frac{\partial P_0}{\partial n_0} \right)_{\frac{s_0}{n_0}} .
\tag{200}
$$

This is the expected result from the velocity addition formula for a Galilean boost invariant system.

For a Lorentz invariant system we have $\rho_0 = \frac{\tilde{\mathscr{E}}_0 + P_0}{1-v_0^2}$ and $n_0 = \frac{\tilde{n}_0}{(1-v_0^2)^{1/2}}$ with $P_0 = P_0(\tilde{\mathscr{E}}_0, \tilde{n}_0)$. This leads to $V = (1-v_0^2)X$, $Y = X$ and $Z = \frac{1}{1-v_0^2}(X-1)$ where $X = \tilde{v}_s^2$ with

$$
\tilde{v}_s^2 = \left( \frac{\partial P_0}{\partial \tilde{\mathscr{E}}_0} \right)_{\tilde{n}_0} + \frac{\tilde{n}_0}{\hat{\mathscr{E}}_0 + P_0} \left( \frac{\partial P_0}{\partial \tilde{n}_0} \right)_{\tilde{\mathscr{E}}_0} .
\tag{201}
$$

It can be shown by squaring $v_s + \left( \frac{X+Y}{2(1-v_0^2 Z)} - 1 \right) v_0 \cos \theta$ and using the expressions for $V, X, Y, Z$ that we have

$$
\left( \frac{1}{\tilde{v}_s^2} - 1 \right) \frac{1}{1-v_0^2} (v_s - v_0 \cos \theta)^2 + v_s^2 - 1 = 0 .
\tag{202}
$$

Let us define $(\tilde{\omega}, \tilde{k}^i)$ so that $\tilde{v}_s^2 = \frac{\tilde{\omega}^2}{\tilde{k}^2}$. Then, using $v_s^2 = \frac{\omega^2}{k^2}$, we obtain

$$
\left( \tilde{k}^2 - \tilde{\omega}^2 \right) \frac{1}{\tilde{\omega}^2} \frac{1}{1-v_0^2} (\omega - v_0 k \cos \theta)^2 + \omega^2 - k^2 = 0 .
\tag{203}
$$

This can be written as the Lorentz invariant statement

$$
\tilde{\omega}^2 - \tilde{k}^2 = \omega^2 - k^2 ,
\tag{204}
$$

if we define

$$
\tilde{\omega} = \frac{1}{(1-v_0^2)^{1/2}} (\omega - v_0 k \cos \theta) = \frac{1}{(1-v_0^2)^{1/2}} \left( \omega - v_0^i k^i \right) .
\tag{205}
$$

We conclude that $(\tilde{\omega}, \tilde{k}^i)$ are related to the LAB frame frequency and momentum $(\omega, k^i)$ by a Lorentz transformation with parameter $v_0^i$. This means that $\tilde{v}_s^2$ is the speed of sound in the comoving frame. The frequencies $\tilde{\omega}$ and $\omega$ are related by the relativistic Doppler effect.

For a Lifshitz scale invariant system we have $P_0 = \frac{z}{d} \tilde{\mathscr{E}}_0 + \frac{z-1}{d} \rho_0 v_0^2$. This implies the following two relations among $V, X, Y, Z$,

$$
V = \frac{z}{d} \frac{\tilde{\mathscr{E}}_0 + P_0}{\rho_0} + \frac{z-1}{d} v_0^2 (X-1) ,
\tag{206}
$$

$$
Y = -\frac{z-2}{d} + \frac{z-1}{d} v_0^2 Z .
\tag{207}
$$

In the LAB frame when $v_0^i k^i = 0$ the speed of sound $v_s^2 = \frac{\omega^2}{k^2}$ is given by

$$v_s^2 = \frac{z}{d} \frac{\tilde{\mathscr{E}}_0 + P_0}{\rho_0} - \frac{z-1}{d} v_0^2 + \frac{1}{d} \frac{v_0^2}{1 - v_0^2 Z} X \,. \tag{208}$$

When $v_0^i k^i = v_0 k$ we have on the other hand,

$$
\begin{aligned}
0 \;=\; & (1 - v_0^2 Z)(v_s - v_0)^2 + \left( -\frac{z-1}{d}(1 - v_0^2 Z) + \frac{1}{d} + X \right) v_0 (v_s - v_0) \\
& - \frac{z}{d} \frac{\tilde{\mathscr{E}}_0 + P_0}{\rho_0} - \frac{z-1}{d} v_0^2 (X - 1) \,.
\end{aligned}
\tag{209}
$$

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
