# Peer review of "Perfect Fluids"

_SciPost Physics, doi:SciPost Phys. 5, 003 (2018)_

## Round 2 · Referee Report · Anonymous (Referee 1) · 2018-2-10

Strengths

See report below.

Weaknesses

See report below.

Report

I have written several papers in the past few years using hydrodynamics without any kind of boost symmetry, and I view it as a more or less trivial generalization of "standard" hydrodynamics -- just with less structure and hence with more independent transport coefficients and thermodynamic susceptibilities.

The claim in the abstract of this paper to develop a "new theory of perfect fluids ... in the absence of any kind of boost symmetry" therefore made me somewhat skeptical. Unfortunately this skepticism did not diminish as I starting reading the paper.

Most of the conceptual structure developed at the start of this paper can be found in section 5.4.5 (and other sections) of the review https://arxiv.org/abs/1612.07324. In particular, the "new fluid variable" defined in equation 2.8 of the paper under review is the same as that appearing in equation 459 of the aforementioned review. This quantity is referred to as M or as chi_PP in many previous works. The inclusion of v.dp terms in discussion of thermodynamic variations can also be found there and in other papers (often rather briefly, the question is to what extent a more elaborate exposition is necessary). The sound speed -- the second "main result" of the paper under review -- will follow immediately from the hydrodynamical constitutive relations, conservations laws, and thermodynamic relations. The consequences of scaling ward identities on thermodynamic variables is also rather simple and previously discussed in several places (including the review above).

Part of the problem of the paper is that it seeks to establish a contrast (in e.g. the first few pages) with the way hydrodynamics is usually set up, supposedly built around Galilean or Lorentzian boosts. However, this is not really the right way to set up hydrodynamics even in those cases -- the basic symmetries are translations that guarantee the existence of the hydrodynamical velocity field. They authors may find useful the development of hydrodynamics in e.g. the textbook by Chaikin and Lubensky. While that book does restrict to Galilean-invariant systems for the most part, the boost symmetry is not instrumental in the logic, but just sets certain transport coefficients to zero.

The above said, there are fairly widespread misconceptions in both the high energy and condensed matter communities regarding the necessity of (respectively) Lorentzian or Galilean symmetries for hydrodynamics, and this paper is somewhat symptomatic of these misconceptions, even while trying to confront them. It is also true that there is not much work done on showing how e.g. Lorentzian hydrodynamics emerges from a more general structure, especially at the nonlinear level. These are questions that the paper under review addresses to some extent.

Therefore, this could be a useful paper, at the very least sociologically. I think, though, in order for it to be useful it should be fairly majorly re-written. The are two main aspects of this. Firstly, the overall framing should be more modest and contextualized by previous work, explaining to and reminding the reader that hydrodynamics just requires enough symmetries to guarantee the existence of the hydrodynamic modes and not more (as is well known by at least some people, and in some textbooks). Secondly, many of the derivations involve rather elementary thermodynamic etc. manipulations that are spelt out in some detail and applied to several different cases. Much of this could be relegated to appendices, so that the text could bring out the more conceptual way in which certain specific systems, with additional symmetries beyond the bare ones, fit into the more general structure. (Tangentially, there is also a large literature on which symmetries create new hydrodynamic modes and which ones don't -- this shows up especially in the context of symmetry breaking, e.g. why when you spontaneously break translations and rotations simultaneously you don't get goldstones for rotations, only for translations. This brings out forcefully the difference between the role of e.g. translational and boost symmetries).

Once this is done, this will not be an especially innovative paper but it will be one that helps to clarify the nature of hydrodynamics and may well help to reduce confusion in several fields.

Requested changes

See report.

  • validity: high
  • significance: ok
  • originality: ok
  • clarity: good
  • formatting: good
  • grammar: good

Author:  Stefan Vandoren  on 2018-03-12  [id 228]

(in reply to Report 1 on 2018-02-10)

We have been processing the referee reports and would like to make the following comments.

1. As for Referee 1, we would like to stress that because of the absence of boost symmetry, different inertial observers will measure different values for the hydrodynamical observables and quantities. That is one of the reasons why one should compute all the velocity dependent terms in the partition function. Referee 2 clearly points this out, in reply to Referee 1. The relevance of this is also demonstrated in the calculation of the speed of sound for Lifshitz gases. (Applying the wrong formula for Lifshitz sound speeds has appeared in many other papers, leading to the wrong formula $c_s^2=z/d$.)
2. To Referee 1, it is true that the concept of ‘kinetic mass density’ has appeared before, such as in the review 1612.07324. We are happy to include that in our references. But, it is not stressed there that this quantity depends on the frame of the observer. In fact, this quantity only appears in the literature at zero background velocity. Furthermore, as a general comment, that same review states explicitly (see page 97): “…we do not have a systematic understanding of how this arises. To the best of our knowledge, there has not been a systematic development of hydrodynamics for theories that are neither Galilei nor Lorentz invariant.” This is precisely what we started in our paper. If Referee 1 wants us to change “new theory” into “systematic development”, that is fine with us.
3. Writing papers for sociological purposes may sometimes be useful, but is not our primary goal. There is enough new material in the paper, as stated also by Referee 2. It is correct that our paper could have been written decades ago, but it never was.
4. Referee 1 ticks the boxes “High Validity”, “Good Clarity” and “Good Formatting”, yet he/she asks us to majorly rewrite the paper. In our opinion, the paper is well written and accessible to a broad audience, and the authors are experienced enough to give a clear exposition. Moreover, Referee 2 also wants us to rewrite stuff, but wants us to emphasize different things, in tension with what Referee 1 wants, as far as we understand. We believe authors should have the freedom about how they want to present their results and what should be emphasized, as long as it is clear and meeting the scientific standards of the journal. Perhaps the calculation about the speed of sound in a non-zero velocity dependent background could be put to an appendix, we did discuss that before among the authors.
5. As for Referee 2, in our opinion Section 2.5, 2.6 and 2.7 are as important as 2.8. One can argue about 2.4 but it is all subjective and we chose it to do this way.
6. Referee 2 also states that Section 4 contains nothing new or interesting, so please point out for us e.g. where the correct sound speed for Lifshitz gases appears. It is discussed in many places in the literature, and they are all wrong.

Concretely, our proposal is to make the following changes:

1. Rephrase ‘new theory’ into ‘systematic development’ throughout the paper. We keep emphasizing though that computing velocity dependence is important and is one of the new results of the paper. For instance, besides the speed of sound, the no-go theorem about no Schrodinger perfect fluids for z\neq 2 is a nice application of studying velocity dependence, and also a new result. Furthermore, we add some references to the review mentioned above.
2. We will move the calculation of the speed of sound in a non-zero velocity dependent background in an appendix, and some details of Section 4 as well if insisted.
3. We will emphasize a bit more the velocity dependence of the pressure, following the request of Referee 2. It was actually needed in Section 4, so it is easy to make that a bit more visible. All the velocity dependence in the grand partition function is written in section 4, so one can elaborate a bit more on the hydrodynamic quantities here.

Hereby, we are trying to look for a compromise. We do not see much added value by a major rewriting of the draft, as we then rather withdraw. Of course, if the editors/referees agree, we will prepare the file and submit it again for approval.

---

## Round 2 · Referee Report · Anonymous (Referee 2) · 2018-2-22

Strengths

I think this paper has the potential to provide a modern treatment of hydrodynamics without Lorentz/Galilean invariance -- see report.

Weaknesses

I think the presentation at the moment is rather poor and will not get across the main points -- see other referee's report, and my report.

Report

I agree, more or less, with most of the comments of the prior referee, so I will not repeat them. My one point of disagreement is that I think 1612.07324 does not fully solve the hydrodynamic problem of interest here, and so there is a role for a paper of this form to exist -- and to have genuinely new physics in it.

However, the manuscript at present falls very far short of what it could be. I think some of what follows is subjective, so I would be willing to consider the response of the authors. But, my view is this: the way that fluids are introduced in this paper is essentially the way it would have been done in 1960 (if not earlier). It is a shame that the modern way of introducing fluid dynamics is relegated to a short section 2.8. Rather than painfully detailing the differences between Lorentz/Galilean/Carollian/etc. invariances...the authors really should introduce the geometric perspective much sooner and simply explain how the stronger boost symmetries cause the partition function to be more constrained than it would have otherwise been. I think an expanded geometric discussion section 2.8 should form the heart of section 2 -- almost all of the rest of section 2, in particular 2.4 and possibly 2.5-7, should be relegated to appendices. The current presentation makes the paper bulky and buries the interesting new perspectives that this paper gives.

In my view, most of Sections 3 and 4 are not particularly new, or interesting -- Section 3 for example could be put as an appendix. What the authors should do in section 4, in my view, is to remove a lot of the unnecessary clutter about the Galilean/Lorentzian limits, and to focus on a simple point. The non-trivial thing about the hydrodynamics in the absence of boosts -- which was not carefully understood/explained in 1612.07324 -- is the way that the pressure depends on the velocity at the nonlinear level. Besides the elegant geometric argument of section 2.8, the punchline of this paper should be an explanation/demonstration of whether or not the pressure can obtain interesting and subtle v^2 dependence. The authors have certainly put in some work along these lines but, as in section 2, there is just so much calculation about the z=1,2 limits that the interesting new physics feels completely buried. I also feel that the velocity dependence of pressure for general z is not properly unpacked at present. For example, in the limit of low velocities, relativistic and Galilean hydrodynamics can look "similar" (see, e.g. 0809.4512). Can one make a similar statement for general z? What is the velocity dependence of pressure at large z? These are the obvious questions one would ask that go beyond the textbook fluid theory, and this paper needs to confront them directly. This perspective is what would make this paper stand out from what has been understood for a long time.

Requested changes

See report.

---

## Round 3 · Referee Report · Anonymous (Referee 3) · 2018-6-3

Strengths

See report

Weaknesses

See report

Report

My view is that the authors in the end took a large amount of the advice given by the referees including myself previously, in particular reducing a lot the unnecessary clutter in Sections 3 and 4. The paper is now more streamlined and is easier to follow.

(I did not say that there was absolutely nothing new in Section 4. In fact the authors ended up doing mostly what I wanted, which was to focus on the z\ne 1,2 limits in a more streamlined section.)

I had a typo in my earlier review which was that I wrote to talk about "large z" limit but that should have been "large v" limit, although I suppose both are worthwhile -- since this is clearly my "fault" I would not bother the authors to add this to the paper unless they felt compelled to.

I think the paper can be published as is.

  • validity: high
  • significance: good
  • originality: good
  • clarity: high
  • formatting: excellent
  • grammar: perfect

Anonymous on 2018-06-06  [id 266]

(in reply to Report 1 on 2018-06-03)

Thank you for the positive recommendation.

---

## Round 3 · Referee Report · Anonymous (Referee 4) · 2018-6-5

Strengths

The manuscript is well written and for the most part easy to digest.

Weaknesses

The distinction between known results and new results is not made clearly enough.

Report

The manuscript reminds me of an American Journal of Physics style article. I suspect the content is for the most part not new, but fills what appears to be a pedagogical gap in the literature on hydrodynamics in systems without boost symmetry. The authors treat in detail the ideal case, i.e. the case without dissipation. (A fuller treatment is promised in refs. [3] and [4].) The new'' results surround the dependence of hydrodynamic quantities on the kinetic mass density''. The authors compute the speed of sound in their framework, and also provide a no go'' theorem for fluids with Schrodinger symmetry. They also discuss in detail a simple example -- an ideal gas of Lifshitz particles.

Regarding the newness of (2.10), I would point out that a similar quantity shows up in the hydrodynamics of superfluids, where v is replaced by the gradient of the phase of the order parameter. I wonder if (2.10) may show up more generally in the hydrodynamics of two-component fluids.

Requested changes

1) On page 3, mention is made of various places in the literature''. It would be nice to know what these places are.

2) On page 21, there is a similar issue with the statement We give references below''. There are a handful of references to older papers in what follows, but too much work is left to the reader. It would be nice to have a couple of specific sentences outlining what is new here and what is drawn from the literature, with equation numbers.

3) I noticed a handful of typos, indepenent'' on p 11, one can be build'' on p 17.

  • validity: high
  • significance: good
  • originality: good
  • clarity: top
  • formatting: excellent
  • grammar: excellent

Anonymous on 2018-06-06  [id 267]

(in reply to Report 2 on 2018-06-05)

We find the comments of this referee difficult to place in context. Firstly, whether this manuscript reminds the referee of some other journal style, we do not find a very useful comment that we can do something with. Secondly, the referee has a suspicion that most part is not new, but we do not know how to respond to 'suspicion'. The comment about the "newness" of equation (2.10) is a mystery to us, as the velocity is not the gradient of the phase of an order parameter. We do agree though that is interesting to apply our framework to superfluids, as we mention in the discussion and outlook.

As for the requested changes, we believe we have stressed very clearly and explicitly in section 4 what are for sure the new equations, namely the velocity dependent terms in the partition function and thermodynamic quantities, and the speed of sound. We have given the relevant references [32,33] and connected them to e.g. the relevant equations. Some results where known for z=2, but the other referee asked us to take out this special case. Still reference [33] is mentioned connected to eqn (4.45). Whether e.g. the formula for the heat capacity in eqn (4.12) is new, we don't know and we make no strong claims there. We have not found it anywhere in the literature, but agree it is a straightforward generalisation, as we write in the beginning of section 4.

We would be happy to correct the typos the referee mentions in a later stage.

---

## Round 3 · Referee Report · Anonymous (Referee 5) · 2018-6-14

Strengths

See below

Weaknesses

See below

Report

The authors have improved the paper, but have not taken to heart the comments of, at this point, three referees. The comments of these referees are all broadly in agreement and relate to the novelty of the ideas presented. That said, novel or not, the technical content of the paper appears to be correct and may be helpful to some readers. Therefore, I think that it is publishable at this point.

Perhaps I can best illustrate the reservation as follows. The authors emphasize that they are treating the velocity as a chemical potential. In fact, I would have said that is the definition of the velocity (as it appears in e.g. their 2.1), and it's certainly not new. Given this fact, it is obvious that in the absence of boost symmetries the pressure will depend on the velocity. Relatedly, while it's conceivable that 2.10 has never been written down in this form (as the authors claim somewhat boldly in footnote 3 -- have they checked every paper in existence?), in any case it's a trivial consequence of the above fact. On this point, and this just happens to be one paper that I know off the top of my head, the authors may find equation 9 of https://arxiv.org/abs/0809.4870 instructive. Chi in that equation is the superfluid velocity squared, which is indeed not associated to a boost symmetry, and so this equation has a rather strong analogy to 2.9 in the paper under review. Obviously the equations are not the same equation, one has to do with superfluids and the other doesn't, but the point is that once thermodynamics can depend on a velocity this kind of relation is a rather immediate thing that drops out.

Perhaps I would mind less if the authors were able to present these points without claiming novelty at every turn. The word "new" appears 16 times in the paper. Some journals have a policy of not allowing explicit use of "new" and "novel" etc., and perhaps that would help here.

---

## Round 3 · List of Changes

Changes/Modifications in the Draft:

1) Abstract/Intro: changed the phrasing “develop a new theory” into “present a systematic or unified analysis/treatment/framework”.
2) Added in intro and in section 2 a few sentences mentioning that the notion of kinetic mass density was already discussed in the literature (e.g. review 1612.07324), but at the same time stress that this was at zero velocity (lab frame), whereas we want to know it in all frames. Moreover, we stressed in the intro that this kinetic mass density follows from the pressure at finite velocity. Therefore, it is important to determine velocity dependent terms.
3) Streamlined in the intro better the three main new results obtained in this paper, stressing also the unified picture we give that can be applied to all perfect fluids with and without boost symmetry.
4) Restructured Section 3, and moved previous section 3.2 (boosted fluids) to an appendix.
5) Major editing in Section 4: deleted all the discussion of the special values of z=1 and z=2, such that the exposition is shorter and flows more smoothly. Furthermore, we added some more text about the velocity dependent terms, for instance in the pressure.
6) Added a discussion about the large z-limit, and the velocity dependent terms in that limit, also in Section 4.

---

## Editorial Decision

published